# Intrinsic magnetic topological insulator phases in the Sb doped MnBi$_2$Te$_4$ bulks and thin flakes

Bo Chen[1,2,9], Fucong Fei[1,2,9]*, Dongqin Zhang[1,9], Bo Zhang[3,9], Wanling Liu[4,5,6,9], Shuai Zhang[1,2], Pengdong Wang[3], Boyuan Wei[1,2], Yong Zhang[1,2], Zewen Zuo[1,2], Jingwen Guo[1,2], Qianqian Liu[1,2], Zilu Wang[7], Xuchuan Wu[7], Junyu Zong[1], Xuedong Xie[1], Wang Chen[1], Zhe Sun[3], Shancai Wang[7], Yi Zhang [1], Minhao Zhang[1,2], Xuefeng Wang[2,8], Fengqi Song[1,2]*, Haijun Zhang[1]*, Dawei Shen [5,6]* & Baigeng Wang[1]

Magnetic topological insulators (MTIs) offer a combination of topologically nontrivial characteristics and magnetic order and show promise in terms of potentially interesting physical phenomena such as the quantum anomalous Hall (QAH) effect and topological axion insulating states. However, the understanding of their properties and potential applications have been limited due to a lack of suitable candidates for MTIs. Here, we grow two-dimensional single crystals of Mn(Sb$_x$Bi$_{(1-x)}$)$_2$Te$_4$ bulk and exfoliate them into thin flakes in order to search for intrinsic MTIs. We perform angle-resolved photoemission spectroscopy, low-temperature transport measurements, and first-principles calculations to investigate the band structure, transport properties, and magnetism of this family of materials, as well as the evolution of their topological properties. We find that there exists an optimized MTI zone in the Mn(Sb$_x$Bi$_{(1-x)}$)$_2$Te$_4$ phase diagram, which could possibly host a high-temperature QAH phase, offering a promising avenue for new device applications.

[1] National Laboratory of Solid State Microstructures, Collaborative Innovation Center of Advanced Microstructures, and College of Physics, Nanjing University, 210093 Nanjing, China. [2] Atomic Manufacture Institute (AMI), 211805 Nanjing, China. [3] National Synchrotron Radiation Laboratory, University of Science and Technology of China, 230029 Hefei, China. [4] Division of Photon Science and Condensed Matter Physics, School of Physical Science and Technology, ShanghaiTech University, 200031 Shanghai, China. [5] State Key Laboratory of Functional Materials for Informatics, Shanghai Institute of Microsystem and Information Technology (SIMIT), Chinese Academy of Sciences, 200050 Shanghai, China. [6] Center of Materials Science and Optoelectronics Engineering, University of Chinese Academy of Sciences, 100049 Beijing, China. [7] Department of Physics and Beijing Key Laboratory of Opto-electronic Functional Materials & Micro-nano Devices, Renmin University of China, 100872 Beijing, China. [8] National Laboratory of Solid State Microstructures, Collaborative Innovation Center of Advanced Microstructures, and School of Electronic Science and Engineering, Nanjing University, 210093 Nanjing, China. [9] These authors contributed equally: Bo Chen, Fucong Fei, Dongqin Zhang, Bo Zhang, Wanling Liu. *email: feifucong@nju.edu.cn; songfengqi@nju.edu.cn; zhanghj@nju.edu.cn; dwshen@mail.sim.ac.cn

Recently, research on topological phases has created a surge of interest and some fruitful outcomes for condensed matter physics and material science. Different combinations of topology and magnetism reveal more interesting topological phases, in which various quasiparticles with or without counterpart particles in the universe can be simulated in condensed matter; for example, quantum anomalous Hall states, topological superconducting states, topological axion insulating states, and Weyl semimetallic states[1–12]. Despite intense efforts, this research is still in its early stages due to obstacles in the material platforms. In particular, magnetic topological insulators (MTIs), which are expected to reveal both quantum anomalous Hall effects (QAHEs) and topological axion insulating states, are extremely rare. Previous efforts have been made to study the doping of magnetic impurities in topological materials or the proximity effect in magnetic heterostructures of topological materials and magnetic insulators[13–21], in which the magnetic effect is weak. Generally, increased levels of dopants may increase the exchange field but can reduce sample quality and decrease electronic mobility[22–29]. Therefore, most exotic topological quantum states including QAHE are either only observable at extremely low temperatures or have yet to be observed experimentally[13,16,27]. These drawbacks have dragged behind the pace of development of these materials for practical applications. Hence it is a key priority to search for more intrinsic MTIs, with less stringent requirements for the quantum effects mentioned above. The recently discovered $MnBi_2Te_4$ is found to be a possible intrinsic MTI with spontaneous antiferromagnetic (AFM) magnetization[30–32]. Theoretical calculations show that QAHE states may arise in $MnBi_2Te_4$ thin film and that the QAHE gap is almost 50–80 meV, which is much larger than that of magnetically doped topological insulators[30,33], pointing to the possibility that QAHE could be observed at elevated temperatures. $MnBi_2Te_4$ also offers a promising platform for topological axion insulators and Weyl semimetals[30,31]. However, according to recent reports of angle-resolved photoemission spectroscopy (ARPES)[34–38], synthesized crystals of $MnBi_2Te_4$ always show heavy n-type doping, and the Fermi level lies in the bulk conduction bands[34–38]. Thus, for undoped $MnBi_2Te_4$ with bulk carrier dominated transport, there is less likely to be evidence of exotic magnetic topological properties, including QAHE.

In this work, we grow a serial single crystals of the $Mn(Sb_xBi_{(1-x)})_2Te_4$ family, and search for intrinsic MTIs in this family. By combining ARPES, low-temperature transport measurements, and first-principles calculations, we reveal several physical transitional processes, including n–p carrier transition, magnetic transition, and topological phase transition in this family of materials. We reveal that the Fermi level of $Mn(Sb_xBi_{(1-x)})_2Te_4$ can be tuned from the conduction bands to the valence bands by adjusting the atomic ratio x. Specifically, for bulk crystals, the Fermi level lies near the bulk gap when x ~ 0.3, the sample with x ~ 0.3 displays the highest resistivity, and an n–p carrier transition occurs near this atomic ratio. For thin-film devices, less substitution of antimony, with x ~ 0.1, is required because chemical potential shifting occurs during device fabrication. Theoretical calculations reveal that the inverted bandgap in $Mn(Sb_xBi_{(1-x)})_2Te_4$ holds when x < 0.55. We find an optimized zone in the $Mn(Sb_xBi_{(1-x)})_2Te_4$ phase diagram, thereby achieving an intrinsic MTI combining topological non-triviality, spontaneous magnetization, and bulk carrier suppression. This provides an ideal platform for the realization of novel topological phases such as high-temperature QAHE and topological axion states.

## Results

### Topological nontrivial states in the crystals at x = 0. We investigate the whole family of $Mn(Sb_xBi_{(1-x)})_2Te_4$ materials,

starting from crystals with x = 0. They have a layered rhombohedral crystal structure with the space group $R\bar{3}m$[39], composed of stacking septuple layers (SLs) Te–X–Te–Mn–Te–X–Te (X = Bi/Sb) as displayed in Fig. 1a. The magnetic moments of each manganese atom in a SL are in parallel with each other and form a ferromagnetic (FM) order with the out-of-plane easy axis, while the magnetizations of the neighboring SLs are the opposite, forming an AFM order[34,35]. Single crystals of $MnBi_2Te_4$ are grown using the flux method. Plate-like single crystals with sizes larger than $5 \times 5$ mm$^2$ are obtained, which can easily be exfoliated mechanically along the c axis. The cleavage surfaces are flat and shiny, and suitable for ARPES measurements.

We conduct ARPES measurements on the x = 0 samples. Figure 1b shows the ARPES intensity plot around the center of the Brillouin zone, with a photon energy of 7.25 eV at 8 K. In this plot, one can see the parabolic bulk conduction band (BCB) and valence band (BVB), of which the dispersions are broadened and unclear due to bulk projections (marked by the orange arrows). In addition, a clearer Dirac-cone-like feature with linear dispersions can also be seen in the bulk bandgap (marked by the red arrows), with a crossing point at $k_\parallel = 0$ and a binding energy of −0.27 eV, which can be identified as the topological surface states (TSSs). Considering the range of the Dirac-cone-like dispersions, as shown in the magnified view in Fig. 1c and the corresponding constant energy contours in Fig. 1d, the isotropic circular Dirac-cone shaped dispersion formed by the two linear TSS bands can be seen more clearly. Although an exchange gap should open at the Dirac point[30,31,33], we note that the linear dispersion of these two bands seems well maintained at the crossing point and no observable gap opening feature can be observed. According to the momentum distribution curves (MDCs) displayed in Fig. 1e, there is still a finite spectral weight near the Dirac point (bold line), and the intensity of the energy distribution curve (EDC) near the Dirac point changes almost linearly with no obvious 'U' shaped gap opening features in Fig. 1f. Considering the resolution limit of our ARPES, we believe that the gap opening size in real synthesized $MnBi_2Te_4$ crystals is much smaller (<25 meV) than calculations (~80 meV)[33] and cannot be identified by our ARPES measurements, the smaller value being consistent with the transport measurements[40]. The difference between the theoretical calculations and experimental measurements possibly comes from the complex magnetic moment distributions at the surface of the $MnBi_2Te_4$ crystals, where the moments may not be arranged strictly like the A-type AFM order in the bulk. In order to identify further characteristics of bulk band dispersions, we also perform ARPES measurements under various photon energies. We note that the band dispersions of the crystal vary dramatically under different photon energies. Clear surface states can only be detected under low photon energies. When increasing the photon energy, ARPES signals are dominated by the bulk contributions (Fig. 1g). One possible reason for this is the difference between photoemission matrix elements for surface states and bulk states.

### Tuning the n–p carrier compensation by increasing x. In the data presented above, we note that the position of the Fermi level of $MnBi_2Te_4$ is far from the bulk gap. This is because heavy n-type doping is induced during crystal synthesis, and the Fermi level lies in the bulk conduction band. Such an unintentional doping effect also occurs in the samples grown by molecular beam epitaxy and the melting method[35–37], preventing the material (x = 0) from being an ideal MTI for QAHE measurements. Encouraged by some previous successes[41–45], we investigate the whole family of $Mn(Sb_xBi_{(1-x)})_2Te_4$ by increasing the value of x in the search for the ideal bulk-insulating MTI. Shiny

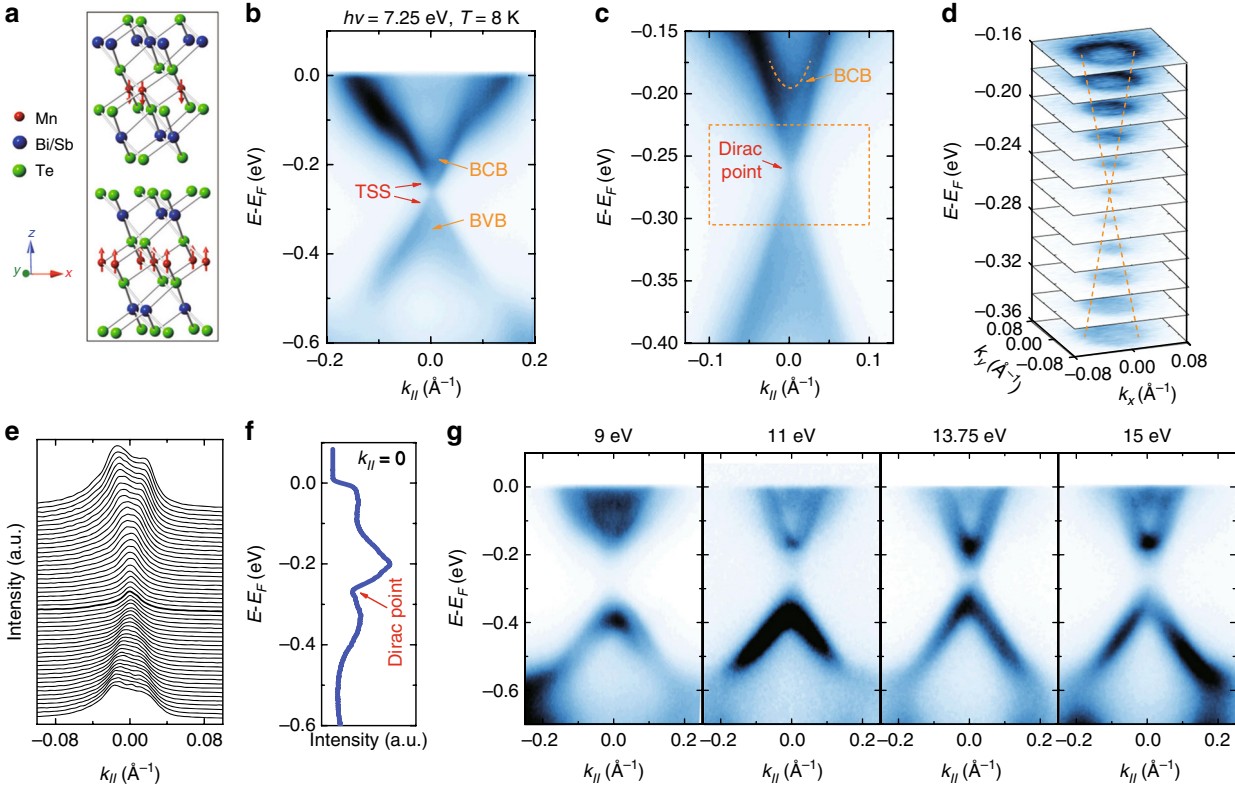

**Fig. 1** ARPES measurements of the crystal with $x = 0$. **a** Crystal structure of the material family. The red arrows indicate the magnetic moment of the manganese atoms. **b**, **c** ARPES data of the crystal and the corresponding magnified view near the Dirac point, under a photon energy of 7.25 eV at 8 K. The arrows in **b**, **c** indicate the bulk conduction band (BCB), bulk valence band (BVB), and the topological surface states (TSSs), respectively. **d** Constant-energy maps at binding energies from −0.16 to −0.36 eV (photon energy = 7.25 eV, $T = 8$ K). **e** Momentum distribution curves derived from the ARPES spectrum in the dashed rectangle area in **c**. **f** The energy distribution curve extracted from the ARPES spectrum in **b** at $k_{||} = 0$. **g** Dispersions of the $x = 0$ crystal under photon energies from 9 to 15 eV

plate-like $Mn(Sb_xBi_{(1-x)})_2Te_4$ crystals could still be obtained following a similar growth process (inset of Fig. 2a). The X-ray diffraction (XRD) patterns for single crystals with corresponding $x$ value are shown in Fig. 2a. The positions of the (00n) diffraction peaks reveal the $c$ axis of ∼ 41 Å in all four samples, consistent with the crystals with $x = 0$. Figure 2b displays the energy dispersive spectra (EDS) for samples with different amounts of antimony substitution. The ratio of Mn:(Bi + Sb):Te is nearly 1:2:4, indicating a stoichiometric atomic ratio of $Mn(Sb_xBi_{(1-x)})_2Te_4$. After normalizing each EDS curve by the amplitude of the peaks from the tellurium, the intensity of the bismuth and antimony peaks clearly evolves as expected, indicating the successful control of the atomic ratio of these two elements.

In order to identify the Fermi levels of the different samples in the $Mn(Sb_xBi_{(1-x)})_2Te_4$ family, we perform ARPES measurements on samples with different antimony ratios. We note that the signals of the antimony-substituted samples are less clear when the photon energy is lower than 10 eV, and following extensive measurements, we choose $hv = 14$ eV for the detections. Even though the ARPES signals at this photon energy are dominated by the bulk contributions, this is sufficient for verifying the evolution of the shifting chemical potential in the different samples. Figures 2c-f show the dispersions detected on the cleaved surface of $Mn(Sb_xBi_{(1-x)})_2Te_4$ for $x$ from 0 to 0.4. The offset of the Fermi level in the different samples is significant. At $x = 0$, the band gap is below the Fermi level at the binding energy of approximately −200 meV, illustrating heavy n-type doping (Fig. 2c). In the sample for $x = 0.2$, the bulk conduction band is still observable in Fig. 2d but the Fermi level is much closer to the band gap than the one shown in Fig. 2c. By increasing the Sb

substitution ratio slightly to $x = 0.3$, a dramatic shift in Fermi level is observed (Fig. 2e). The Fermi level of the samples with $x = 0.3$ lies near the edge of the valence band. The sudden change in Fermi level may be caused by the low carrier density near the bulk gap, and the introduction of a small amount of carrier has a dramatic effect on the chemical potential. When the Sb ratio is increased further to $x = 0.4$, the Fermi level continuously penetrates deep into the valence band, as expected (Fig. 2f). Substituting antimony for bismuth gives rise to the monotonic shifting of the Fermi level from the conduction band to the valence band, indicating an n–p carrier transition in the vicinity of $x \sim 0.3$.

In order to confirm the n–p carrier evolution further, we measure the transport properties of these four samples (bulk crystals with thicknesses of 50–200 μm) at low temperatures. The Hall resistivities $\rho_{xy}$ as a function of the magnetic field $B$ of samples with different substituting ratios when the temperature is higher and lower than $T_N$ are shown in Fig. 2g, h, respectively. The slope of $\rho_{xy}$ changes signs from negative to positive when $x$ increases from 0.25 to 0.3, confirming the n–p transition for the changing Sb ratio, which is consistent with the ARPES measurements. One can further extract the carrier density of each sample by $R_H = 1/ne$, where $R_H$ is the Hall coefficient. Figure 2i displays the calculated carrier density of each sample extracted from Fig. 2h. The carrier density in the sample with $x = 0.3$ is $1.85 \times 10^{18}$ cm$^{-3}$, which is almost 40 times lower than the values for the $x = 0$ sample ($7.05 \times 10^{19}$ cm$^{-3}$), indicating a successful suppression of the bulk carriers by controlling the Sb:Bi ratio in $Mn(Sb_xBi_{(1-x)})_2Te_4$. Bulk carrier suppression can also be identified from the curves of resistivity against temperature for

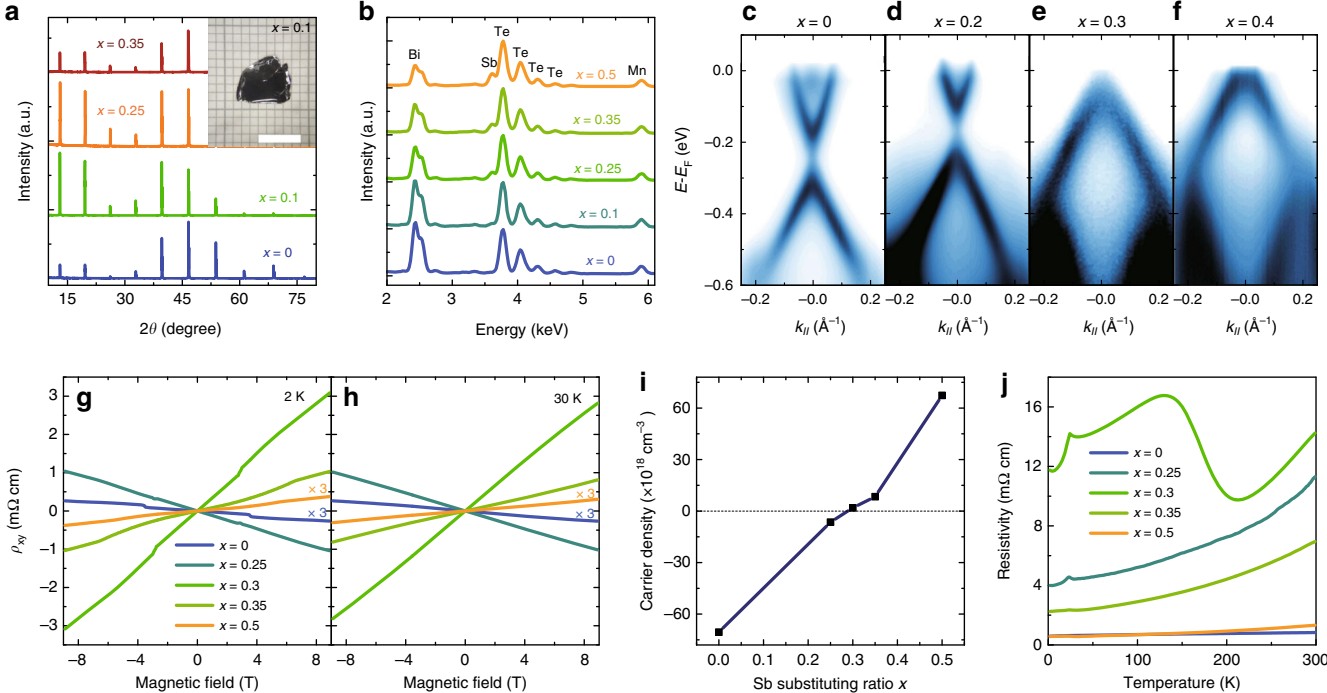

**Fig. 2** Crystal characterization and n–p transition in Mn(Sb$_x$Bi$_{(1-x)}$)$_2$Te$_4$. **a** Single crystal XRD pattern for samples with different antimony substitutions. Scale bar: 5 mm. **b** EDS curves of the five samples for $x = 0$ to 0.5. The curves are offset for clarity. **c–f** ARPES measurements for the samples with different antimony substituting ratio. **g, h** Measurements of Hall resistivity $\rho_{xy}$ as a function of the magnetic field in samples with different values of $x$ at 2 and 30 K, respectively. **i** Corresponding carrier density of samples with different $x$ derived from the Hall resistivity in **e**. **j** Resistivity versus temperature of samples for various values of $x$

samples with different values of $x$ (Fig. 2j). The resistivity of the sample with $x = 0.3$ is highest among all five samples over the whole temperature ranging from 2 to 300 K. In addition, most of the samples show metallic behavior except for the sample of $x = 0.3$, in which the resistivity increases when cooling near 200 K. This increase in resistivity could possibly originate from bulk carrier suppression by fine tuning the ratio of antimony substitution. It may also be a spin-related feature that emerges when the bulk metallic behavior is suppressed, given that considerable spin fluctuation in MnBi$_2$Te$_4$ is reported even at room temperature[36], and the resistivity could be affected. It is reasonable to suppose that there is an electrical neutral point for minimum bulk conductivity contributions, which can be achieved between $x = 0.25$ and 0.35 when the Fermi level aligns with the QAHE gap.

**Magnetism in bulk crystals and thin film devices**. Apart from bulk carrier suppression, another precondition for QAHE is spontaneous magnetization, thus it is necessary to investigate systematically the magnetism of this material family. We use a vibrating sample magnetometer to measure the magnetization of bulk crystals with different values of $x$. Figure 3a shows the field-cooled (FC) and zero-field-cooled (ZFC) curves of four samples for $x = 0$ to 0.5. The FC and ZFC curves of each sample perfectly overlap with each other and a peak near 25 K can be identified, indicating an AFM transition in all four samples from $x = 0$ to 0.5. In the curves of magnetization versus $B$ field displayed in Fig. 3b and the transport magnetoresistances (Fig. 3c) of the samples with different $x$ values (as well as in $\rho_{xy}$ discussed above in Fig. 2g), two types of "kink" feature (marked by the arrows and triangles in Fig. 3b, c, respectively) can be identified near $B = 3$ T and 7 T in all four samples. This can be explained by an AFM to canted-AFM (CAFM) transition ($H_{c1} \sim 3$ T) and a CAFM to FM

transition ($H_{c2} \sim 7$ T) caused by the external magnetic field[36]. One further phenomenon of interest is that when increasing the magnetic field, the magnetoresistance of the sample with $x = 0.25$ jumps upwards at the first kink near $H_{c1} \sim 3$ T, as shown in Fig. 3c, which is the opposite of the case for the other three samples. The reasons for this are worthy of further study and may be attributable to the different magnetic coupling mechanisms between the sample with strong metallic behavior and that with the Fermi level near the band gap[46–48]. Figure 3d summarizes the results of Néel temperature extracted from the magnetization data in Fig. 3a and the corresponding critical field for the kink ($H_{c1}$, $H_{c2}$) in Fig. 3b, c. Both $T_N$ and the critical field decrease with increasing $x$, indicating the weaker AFM coupling in samples with higher $x$. However, the increase in antimony does not much affect the order of AFM because $T_N$ only decreases by 2.6 K in the sample for $x = 0.5$ compared with the one for $x = 0$. We therefore believe that there is no magnetic transition from AFM to paramagnetism and the spontaneous magnetization is maintained in Mn(Sb$_x$Bi$_{1-x}$)$_2$Te$_4$ within the range of $x < 0.5$. In fact, spontaneous AFM order still maintains when $x = 1$, i. e. in pure MnSb$_2$Te$_4$[49].

The synthesized Mn(Sb$_x$Bi$_{(1-x)}$)$_2$Te$_4$ crystals can also be mechanically exfoliated into thin films. The thickness of the thin films is identified by an atomic force microscope (Fig. 4a, b), followed by a standard electron beam lithography process to fabricate the mesoscopic devices (Fig. 4c). Figure 4d shows a high resolution transmission electron microscopy (HR-TEM) photograph and the fast Fourier transformation (FFT) pattern collected on the exfoliated (001) surface of a typical thin film, in which the hexagonal atomic fringes from the (110) lattice plane with a spacing of 0.217 nm are clearly shown, indicating the high quality of the exfoliated thin film crystals. During the transport measurement, we are surprised to discover that the fabrication process of the thin-film device causes the Fermi

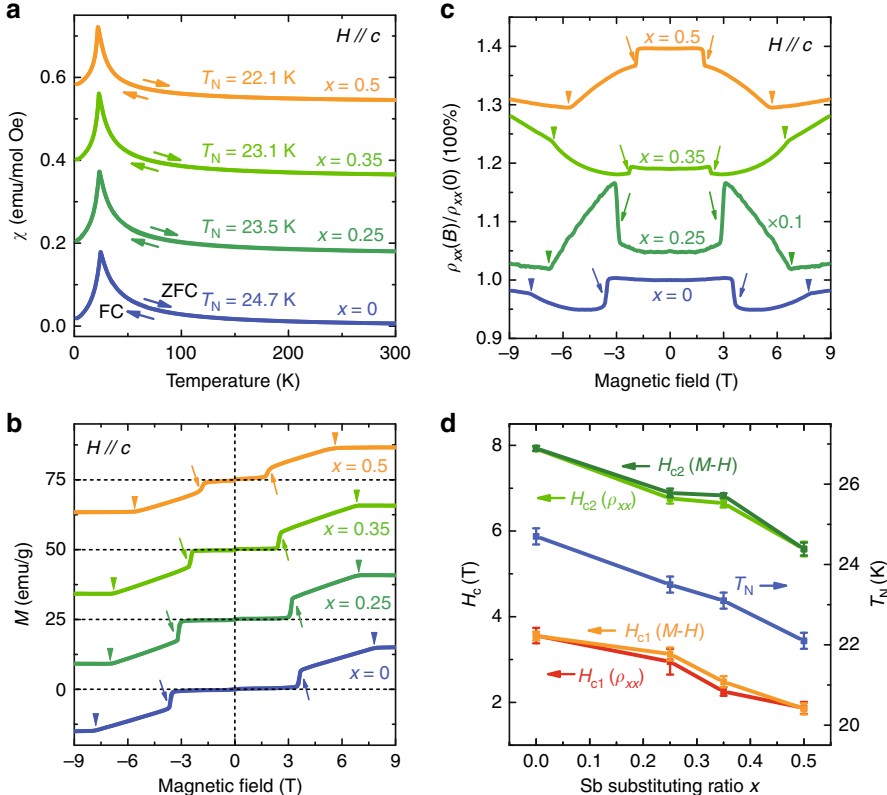

**Fig. 3** Magnetism measurements of $Mn(Sb_xBi_{(1-x)})_2Te_4$. **a** FC-ZFC curves of the samples with different values of $x$. The curves are offset for clarity. **b**, **c** Magnetization versus magnetic field and the magnetoresistance of samples with different values of $x$. The curves are offset for clarity. **d** The evolution of the Néel temperature extracted from panel **a** and the corresponding critical field for the kink ($H_{c1}$, $H_{c2}$) in different samples for various $x$ extracted from **b**, **c**. Error bars correspond to standard errors

level to shift downwards. In other words, the Fermi level of thin films is closer to the Dirac point than bulk crystals for n-type samples. Thus, the optimized $x$ value for thin devices is < 0.3. After a large number of measurements, we find that $x \sim 0.1$ is a more appropriate substitution ratio for thin-film devices (Supplementary Fig. 1). Due to the A-type antiferromagnetism in $Mn(Sb_xBi_{(1-x)})_2Te_4$, the magnetizations of neighboring SLs are opposite, and there should be oscillations in magnetic properties between thin films with even and odd SLs. Figure 4e displays the Hall resistances of several devices with $x = 0.1$ with different thicknesses. Each curve is measured at 2 K under a certain back gate voltage ($V_g$) to reach the maximum longitudinal resistance of the corresponding device. One can see that the devices with odd SLs (5, 7, 9 SLs) show obvious AHE with a coercive field around 0.5 to 1T. Although hysteresis loops can also be observed in samples with even SLs (6, 8 SLs), the coercive fields of these samples are much smaller than the ones with odd SLs and the remnant magnetic signals are believed to result from the canting or disorder of the AFM configurations.[35] Since the Fermi level is close to the charge neutral point, the reversal of AHE signals in Fig. 4e may come from competition between the intrinsic Berry curvature and Dirac-gap enhanced extrinsic skew scattering in this material.[50] Figure 4f summarizes the coercive fields for devices with different thicknesses, and the expected oscillation between even and odd SLs is clearly demonstrated.

**The phase diagram of the multiple transition processes.** As shown above, the bulk carrier contributions of the $Mn(Sb_xBi_{(1-x)})_2Te_4$ family can be suppressed and the crystals have similar

antiferromagnetic interlayer coupling. However, special attention should be focused on the fact that a higher value of $x$ reduces the strength of the spin orbital coupling (SOC) of the materials, which is the origin of the topological transition and the band inversion.

To verify our observations, we calculate the band structures of $Mn(Sb_xBi_{(1-x)})_2Te_4$ using first-principles calculations. Figures 5a–d display the band calculations along with the evolution of the Wannier charge centers (WCCs) in $MnBi_2Te_4$ ($x = 0$) and $MnSb_2Te_4$ ($x = 1$) respectively. According to our calculations, $MnBi_2Te_4$ is an intrinsic MTI with an inverted bulk band gap of $\sim 0.2$ eV, while $MnSb_2Te_4$ is a topologically trivial magnetic insulator with a bulk band gap of 34 meV, indicating a topological phase transition in $Mn(Sb_xBi_{(1-x)})_2Te_4$ when the amount of antimony is increased. Figure 5e shows its phase diagram with the topological phase transition, as well as the n–p carrier transition. The purple curves indicate the calculated bulk bandgap in $Mn(Sb_xBi_{(1-x)})_2Te_4$. The closing and reopening of the bandgap clearly show that the topological transition point lies at $x = 0.55$, which exceeds the value of $x = 0$ to 0.5 in our samples as discussed above. Therefore, our calculations support the fact that the topologically nontrivial characteristics are maintained in the samples considered in this work. For bulk crystals in particular, the electrical neutral point is close to $x \sim 0.3$, while a lower amount of Sb substitution with $x \sim 0.1$ is appropriate in thin-film devices due to the unexpected shift in Fermi level. A peak in the bandgap curve at around $x = 0.9$ can also be observed and we believe this originates from the competition between the SOC effect and the chemical bonding effect (Supplementary Fig. 2).

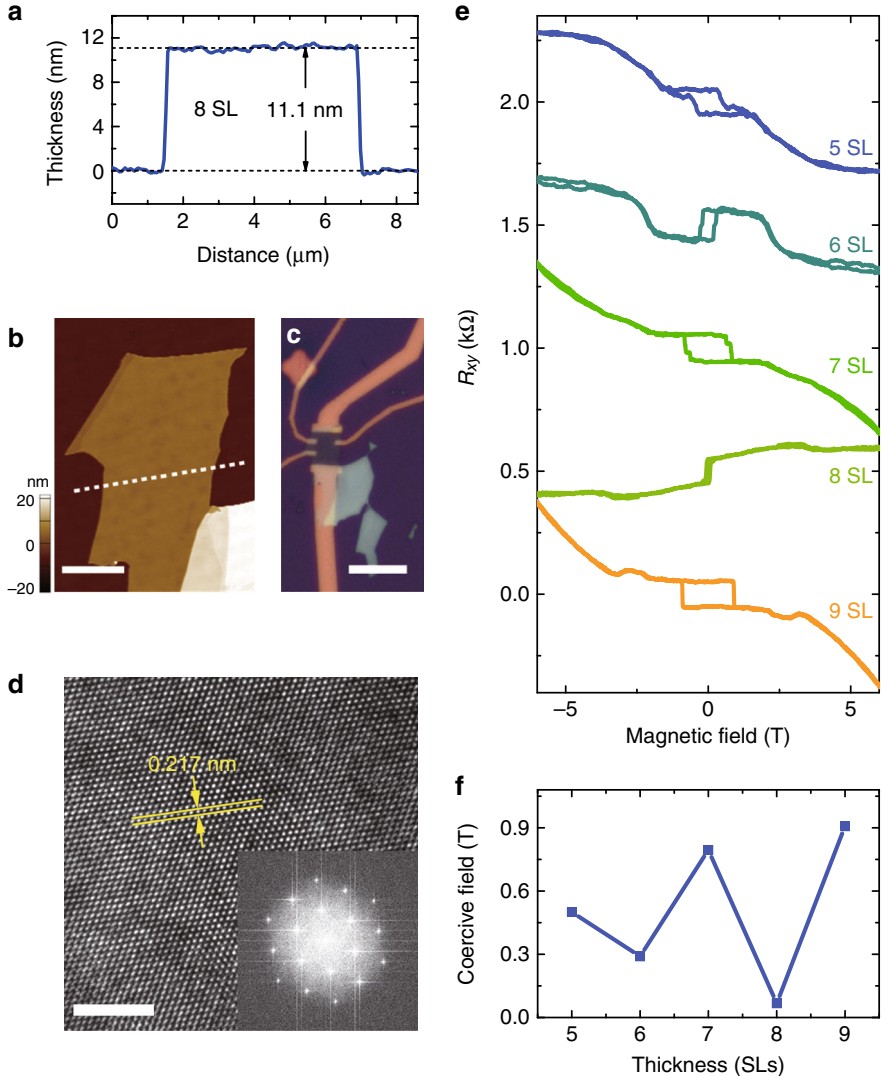

**Fig. 4** Device fabrication and the thickness-dependent AHE signals. **a, b** Atomic force microscopy line scan profile and the corresponding photograph of a typical exfoliated $Mn(Sb_{0.1}Bi_{0.9})_2Te_4$ film with 8 SL. Scale bar in **b**: 3 μm. **c** An optical photograph of the typical fabricated mesoscopic device. Scale bar: 10 μm. **d** An HR-TEM image and the FFT pattern (inset) of a typical exfoliated thin film of $Mn(Sb_{0.1}Bi_{0.9})_2Te_4$. Scale bar: 3 nm. **e** Hall resistance measured at 2 K in the $Mn(Sb_{0.1}Bi_{0.9})_2Te_4$ thin devices with different thicknesses. **f** The thickness-dependent coercive fields extracted from **e**

## Discussion

The three basic requirements for the realization of QAHE, i.e., topological non-triviality, spontaneous magnetization, and bulk carrier suppression, are all satisfied in the sample of Mn $(Sb_xBi_{(1−x)})_2Te_4$ with the optimized substitution ratio $x$ both in bulk crystals and thin films (shown by the dashed ellipse in Fig. 5e), promising an ideal platform for the realization of high-temperature QAHE and other predicted topological effects. In addition, $Mn(Sb_xBi_{(1−x)})_2Te_4$ is a material family hosting rich physical phenomena, including topological phase transition, metallic-insulating transition, n–p type carrier transition, and different magnetisms including AFM, CAFM, and FM. With such potentially physical properties in mind, we believe that this material family is worthy of further study and could offer a wide range of potential applications in the future.

## Methods

**Crystal growth**. High quality single crystals are synthesized using the flux method. For the crystal with $x = 0$, raw materials of MnTe and $Bi_2Te_3$ are mixed in the molar ratio of 1:5.85 in an alumina crucible, which need to be sealed inside a quartz ampule. The ampule is placed in a furnace and heated to 950 °C over a period of one day. After maintaining it at 950 °C for 12 h, the ampule is cooled to 580 °C at a

rate of 10 °C/h. Large-sized $MnBi_2Te_4$ crystals are obtained after centrifuging in order to remove the excess $Bi_2Te_3$ flux. For other crystals with different values of $x$, different amounts of $Sb_2Te_3$ are used to substitute for $Bi_2Te_3$. The growth process is similar but the temperatures for centrifugation differ slightly depending on the ratio of Sb:Bi used. It is noteworthy that the melting temperatures of $MnBi_2Te_4$ and $Bi_2Te_3$ are very close. The actual temperatures of each furnace require careful monitoring and calibration before growth. During the centrifugation, there is a requirement to take the ampule out of the furnace and place it into the centrifuge as soon as possible (normally < 5 s afterwards) because the temperature of the mixture drops fast in an ambient environment.

**ARPES measurements**. ARPES experiments are performed at the BL13U beamline at the Hefei National Synchrotron Radiation Laboratory, and the BL03U beamline at the Shanghai Synchrotron Radiation Facility. Photon energies ranging from 7 to 30 eV are applied in the experiments. The samples are cleaved and measured in an ultrahigh-vacuum chamber at a pressure below $10^{−10}$ Torr.

**Transport measurements on bulk crystals**. The electrical resistivity is measured using a physical property measurement system. (Quantum Design PPMS-14T). The lowest temperature and highest magnetic field are 2 K and 14 T, respectively. The samples used here are plate-like bulk crystals with thicknesses of 50–200 μm and are cleaved by a knife, and the sizes of these plates are 0.5–2 mm. Standard hall-bar contacts are fabricated using silver paste or indium pressing on the cleaved sample surface.

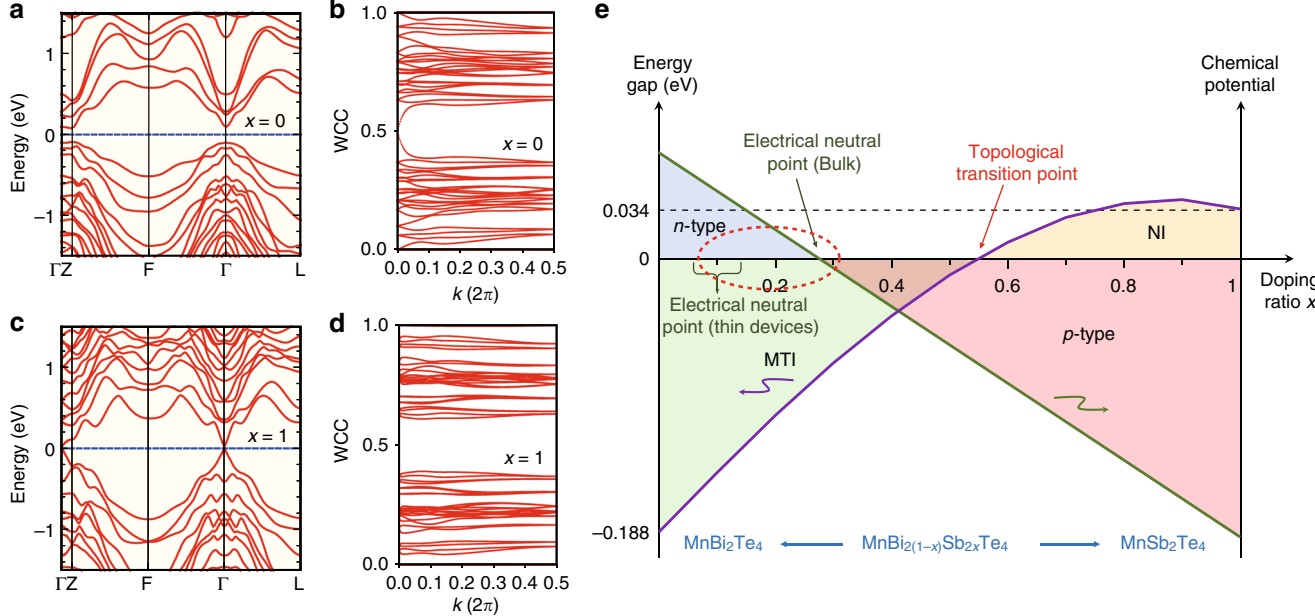

**Fig. 5** Topological properties and the summary phase diagram. **a**, **b** Band structure and the evolution of Wannier charge centers (WCCs) of $MnBi_2Te_4$ ($x = 0$). **c**, **d** Band structure and evolution of WCCs of $MnSb_2Te_4$ ($x = 1$). **e** N–p carrier transition and topological phase transition diagram of $Mn(Sb_xBi_{(1-x)})_2Te_4$

**Magnetic measurements**. A vibrating sample magnetometer equipped in a physical property measurement system (Quantum Design) is used to measure the magnetization of bulk crystals. The measured crystals are about 0.5–1 mm in thickness and 2–3 mm in side length. Standard copper sample holders are used during the measurement. The lowest temperature and highest magnetic field are 2 K and 9 T, respectively.

**Thin device fabrication**. Thin films of $Mn(Sb_xBi_{(1-x)})_2Te_4$ samples are obtained by mechanical exfoliation using scotch-tapes. Silicon wafers with 300 nm silica layers are used as the substrates. An atomic force microscope is used first to determine the film thickness, and thin films with uniform thickness and appropriate size (normally 5–30 μm in length and 3–20 μm in width) are chosen for the next step of device fabrication. Electron beam lithography is used to fabricate Hall-bar electrodes on the thin films.

**First-principles calculations**. First-principles calculations are performed by the Vienna Ab-initio Simulation Package[51] with the local density approximation (LDA) + U functional. Parameters $U = 3$ eV and $J = 0$ eV are used for Mn $d$ orbitals. The virtual crystal approximation[52] is used for $Mn(Sb_xBi_{(1-x)})_2Te_4$. A cut-off energy of 450 eV and $8 \times 8 \times 8$ $k$-points are adopted for self-consistent calculation. Experimental lattice constants are fixed and inner atoms in the unit cell are fully relaxed with a total energy tolerance of $10^{-5}$ eV. The spin-orbit coupling is self-consistently included. Maximally localized Wannier functions[53] are used to calculate the surface states.

## Data availability

The datasets that support the findings of this study are available from the corresponding author upon reasonable request.

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

## Acknowledgements

We gratefully acknowledge the financial support of the National Key R&D Program of China (2017YFA0303203 and 2018YFA0306800), the National Natural Science Foundation of China (91622115, 11522432, 11574217, 11674165, 11774154, 61822403, 11874203, 11904165, 11904166, U1732273, and U1732159), the Natural Science Foundation of Jiangsu Province (BK20160659, BK20190286), the Fundamental Research Funds for the Central Universities, Users with Excellence Program of Hefei Science Center CAS (2019HSC-UE007), and the opening Project of the Wuhan National High Magnetic Field Center. Part of this research used Beamline 03U of the Shanghai Synchron Radiation Facility, which is supported by ME2 project under contract No. 11227902 from National Natural Science Foundation of China. We thank Professor Donglai Feng and Rui Peng from Fudan University for helping with the ARPES measurements. We also thank Professor Ke He from Tsinghua University for some stimulating discussions.

## Author contributions

B.C. and F.F. took charge of crystal growth and characterization. B.C., F.F., B.Z., P.W., W.L., Y.Z., J.G., Q.L., Z.W., X.W., J.Z., X.X., W.C., Z.S., D.S., S.W. and Y.Z. carried out the ARPES measurements. B.C. and F.F. measured the transport and magnetism properties. S.Z. and B.W. fabricated the devices and performed the device measurements. Z.Z. measured the HR-TEM images. D.Z. and H.Z. performed the first-principles calculations. F.F. and F.S. wrote the paper and Y.Z., M.Z., X.W., H.Z. and B.W. participated in discussions on this paper. F.S. was responsible for overall project planning and direction.

## Competing interests

The authors declare no competing interests.
