## [Peer Review File · Nature Communications]

Reviewers' comments:

Reviewer #1 (Remarks to the Author):

This work presents a systematic study on several physical transition processes including n-p carrier transition, magnetic transition, and topological phase transition in the single crystals of $\text{Mn}(\text{Sb}_x\text{Bi}_{1-x})_2\text{Te}_4$. By increasing the Sb substituting ratio, the Fermi level can be tuned from the conduction bands to the valence bands. The authors conclude that there exists an ideal MTI zone ($x=0.3$) in the $\text{Mn}(\text{Sb},\text{Bi})_2\text{Te}_4$ phase diagram. The results are interesting, and the analysis is comprehensive. However, the data is not complete. I feel it is difficult to recommend this paper to be published in NC if the authors could not provide the data for the following comments.

1. How large is the cleaved sample? What's the size of the Hall bar? The authors should supply a TEM image to show the crystal quality, or at least give an AFM or a photograph image of the sample.
2. In Page 10, the authors claim that the abnormal resistivity increase of $x = 0.3$ when cooling near 200 K originates from bulk carrier suppression or the canting of the magnetic moment for the transition from paramagnetism to anti-ferromagnetism. But the Neel temperature of $\text{Mn}(\text{Sb},\text{Bi})_2\text{Te}_4$ is between 20 to 30 K, much lower than 200K. Why the resistivity increase is related to the magnetic transition? The authors should give a more detailed explanation.
3. The authors should present the M-H curve of the sample, and compare it to the anomalous Hall resistance vs. H curve. They should also compare the value of coercive field to the kink point of p_{xx} (H_{c1} and H_{c2}).
4. What's the sample thickness used for the transport and magnetic measurements? As MnBi_2Te_4 is an antiferromagnetic TI composed of ferromagnetic SLs with neighboring SLs coupled antiferromagnetically, there should be oscillation in magnetic properties as the thickness changes between even and odd SLs). Except for the 9 SL and 12 SL, the authors should show more data for the other thickness.

Reviewer #2 (Remarks to the Author):

The authors report on transport properties of magnetic topological insulator (MTI) $\text{Mn}(\text{Sb},\text{Bi})_2\text{Te}_4$ exfoliated single crystal and discuss a topological transition to non-TI by further Sb-doping. The Sb-doping to tune the Fermi level in the bulk band gap is firstly done in this compound. I found the work is technically sound and is an appropriate demonstration for the exploration of an intrinsic MTI phase for MnBi_2Te_4 . The authors may need to address the following concerns before the manuscript could be considered for publication.

- 1) The authors achieved a p-n transition by Sb-doping to MnBi_2Te_4 . According to the theory, the EF-tuned samples with x between 0.25 and 0.35 should show the Axion insulator state or the QAH state depending on the layer number of the sample. Namely, both σ_{xx} and σ_{xy} should be zero in the former case and $\sigma_{xx} = 0$ and σ_{xy} should be quantized to h/e^2 in the latter case. However, the transport data in Figs. 2d and 3b look far from those quantized states. The authors should mention the reason.
- 2) Figure 3d and 3e: The authors shows the presence and absence of hysteresis in R_{yx} depending on the layer numbers of SL. I have questions on the experiment. How homogeneous the layer numbers in the samples? I do not think the sample is completely flat. Then, how the layer numbers are determined? In addition, the authors should provide more detail information about the experiment

such as a typical size, a photograph and the fabrication process of the sample.

3) Related question: what is the layer number of the sample in Figs. 2d-2f? In particular, a ferromagnetic hysteresis is observed?

4) Figure 4e: The size of the band gap changes by increasing Sb due to the reduction of spin-orbit interaction. What is the origin of the peak at $x = 0.9$? Why it shows a non-monotonic behavior?

5) Although the authors claim that a topological phase transition lies at $x = 0.55$ from the band calculation, the samples shown in the manuscript are only $x < 0.5$. They should discuss the phase transition by providing ARPES or transport data for heavily Sb-doped samples.

6) Related to the above, the authors mention in Fig. 3b that "We therefore believe that the magnetic transition point from AFM to paramagnetism in $\text{Mn}(\text{Sb,Bi})_2\text{Te}_4$ lies outside the range studied here". What is the evidence of the magnetic transition to paramagnetism by Sb-doping? The authors should provide a reference or an experimental data.

7) Figure 2b: Because all the EDX data are overlapped and are shown in similar colors, it is quite hard to see the composition dependence. The authors should separate the data into subpanels.

8) Similarly, colors of traces in Figs. 2d and 2f should be revised. In addition, a new figure should be provided that shows a composition x dependence of carrier density.

Reviewer #3 (Remarks to the Author):

The authors studied a series of $\text{Mn}(\text{Sb}_x\text{Bi}_{1-x})_2\text{Te}_4$ samples, by angle-resolved photoemission spectroscopy, transport measurements and DFT calculations. ARPES and transport data show a quite convincing n-p transition, and transport data show AFM magnetic transition in all studied samples ($x \leq 0.5$). Together with DFT calculations, the authors concluded that an ideal MTI zone can be reached by tuning the Sb/Bi ration, where topological nature, bulk carrier suppression, and spontaneous magnetization can be satisfied at the same time, and $x \sim 0.3$ is likely the best composition. These evidences are comprehensive.

However, the main purpose of this study is to find the ideal MTI, and the authors concluded it is $x \sim 0.3$. But they showed almost no data for $x \sim 0.3$. Some of the key evidences (topological surface states and QAH) are carried out only on $x = 0$ or $x = 0.1$ samples, of which similar results were already posted on arXiv a few months ago (The gapped topological surface states for $x = 0$ are also observed in arXiv 1809.07389(Ref.33), 1809.07926(Ref.34), 1812.00339(Ref.35), 1903.11826. The QAH for $x = 0.1$ thin-layer sample is similar to the one reported in arXiv 1809.07926(Ref.34) for $x = 0$). In my opinion, to reduce the overlap with the previous papers and to further support the conclusion, some data on $x \sim 0.3$ (pure topological states, or better QAH, etc) is necessary.

Similar ARPES spectra were already measured by previous papers. Especially, Fig.1 is similar to Fig.3 in arXiv 1809.07389(Ref.33), though experimental quality in Ref.33 seems to be better. The x dependence of ARPES in Fig.2 is new, but spectra are not clear. The ARPES data on $x \sim 0.3$ and its topological surface state is necessary.

There is no clear evidence of topological surface state in ARPES. Spin resolved ARPES is powerful in order to reveal the topological surface state. There have been already similar ARPES data in the

previous papers. The authors should try spin resolved ARPES as a new experiment.

Some comments:

(1) For the ARPES k_z data on $x = 0$ (Fig. 1g), is there any surface state in the second derivative plot? arXiv 1809.07389(Ref.33) observed both bulk bands and surface bands with $h\nu = 9$ eV in the second derivative. If not, what is the possible difference?

(2) According to the calculation (Fig. 1f), the bulk bands show very small k_z dispersion (possibly due to the large lattice constant c). There should be no obvious k_z change for bulk bands in ARPES. In Fig. 1(g), why the data from $h\nu = 13.75$ eV is quite different from others?

(3) In Fig.2(c) panel v ($x = 0.25$), if the red line is the surface band and the black line is the bulk band, the bulk band gap is ~ 250 meV. While for $x = 0$, it is less than 200 meV from Fig. 1(d). The bulk gap should reduce with Sb substitution. How to explain this contradiction?

(4) In Fig.3(d-e), why do the authors use R_{xy} for (d) and ΔR_{xy} for (e)? What is the definition of ΔR_{xy} ?

In summary, there is no strong support for the topological nature and the related QAH for the ideal MTI ($x \sim 0.3$). The data on the chemical potential shift and AF magnetization with Sb substitution are solid. But these solid data are not be enough for Nature Communications.

Reply to Reviewer #1

This work presents a systematic study on several physical transition processes including n-p carrier transition, magnetic transition, and topological phase transition in the single crystals of $Mn(Sb_xBi_{1-x})_2Te_4$. By increasing the Sb substituting ratio, the Fermi level can be tuned from the conduction bands to the valence bands. The authors conclude that there exists an ideal MTI zone ($x=0.3$) in the $Mn(Sb,Bi)_2Te_4$ phase diagram. The results are interesting, and the analysis is comprehensive. However, the data is not complete. I feel it is difficult to recommend this paper to be published in NC if the authors could not provide the data for the following comments.

Reply: First of all, we thank the reviewer for reading our manuscript. We are also grateful to the reviewer for his/her positive comments on our results and analyses. The reviewer also pointed out several concerned issues and advices, which are greatly helpful for us to improve our manuscript. According to these useful advices, we have amended the relevant part in the manuscript. The questions raised by reviewer are answered below in detail.

Comments: *1. How large is the cleaved sample? What's the size of the Hall bar? The authors should supply a TEM image to show the crystal quality, or at least give an AFM or a photograph image of the sample.*

Reply: The reviewer asked the size of our thin film devices and suggested us to provide TEM and AFM images of the sample in the manuscript. We thank the reviewer for the kind suggestion. Our bulk crystals are about $2 \times 2 \text{ mm}^2$ to $5 \times 5 \text{ mm}^2$ mostly. After mechanical exfoliation, the size of the obtained thin films varies from several micrometers to dozens of micrometers. We pick the thin films with uniform thickness (by atomic force microscopy) and appropriate size (normally $5 \sim 30 \mu\text{m}$ in length and $3 \sim 20 \mu\text{m}$ in width) for the next step of device fabrication. Exactly

following the reviewer's request, we provide the AFM photograph of an 8 SL sample in Fig. 4a-b of the revised manuscript (Fig. R1(a)-(b) below). A high resolution TEM image and the fast Fourier transformation pattern of a typical exfoliated thin film are also provided in Fig. 4d of the revised manuscript (Fig. R1(c) below), in which the hexagonal atomic fringes from (110) lattice plane with spacing of 0.217 nm are clearly displayed, indicating the high quality of the exfoliated thin film crystals. An optical photograph of a typical fabricated thin device is also provided in Fig. 4c of the revised manuscript (Fig. R1(d) below).

Fig. R1. (a-b) Atomic force microscopy photograph and the corresponding line scan profile of an exfoliated thin film with 8 SL. (c) A high resolution TEM image and the fast Fourier transformation pattern (inset) of a typical exfoliated thin film of $\text{Mn}(\text{Sb}_{0.1}\text{Bi}_{0.9})_2\text{Te}_4$. (d) An optical photograph of a typical fabricated thin device with thickness of 8 SL.

Comments: 2. In Page 10, the authors claim that the abnormal resistivity increase of $x = 0.3$ when cooling near 200 K originates from bulk carrier suppression or the canting of the magnetic moment for the transition from paramagnetism to

anti-ferromagnetism. But the Neel temperature of Mn(Sb,Bi)₂Te₄ is between 20 to 30 K, much lower than 200K. Why the resistivity increase is related to the magnetic transition? The authors should give a more detailed explanation.

Reply: The reviewer comments that the resistivity increase in the R - T curve of $x = 0.3$ sample cannot result from the magnetic transition. We thank the reviewer for the comments. Here we provide the detailed explanation as follows. For the resistivity increase, we cannot exclude the possibility of some spin related origins because of two reasons. Firstly, according to previous research in some magnetic materials, the increasing of resistivity in R - T curve is reported to result from the influence of magnetic moments, although the temperature is still much higher than the critical temperature¹. Secondly, a recent experimental report points out that the spin fluctuation in MnBi₂Te₄ is still considerable even at the room temperature². Therefore, we cannot rule out the possibility that the resistivity increase is a spin related feature. We apologize that we didn't make a clear explanation on this resistivity increase in the original manuscript. Following the reviewer's suggestion, more explanation is provided in paragraph 1 page 11 of the revised manuscript: "This resistivity increase possibly originates from bulk carrier suppression by fine tuning the ratio of antimony substituting. It also may be a spin related feature emerging when bulk metallic behavior is suppressed, since considerable spin fluctuation in MnBi₂Te₄ is reported even at the room temperature², and the resistivity is possibly influenced."

Comments: 3. *The authors should present the M-H curve of the sample, and compare it to the anomalous Hall resistance vs. H curve. They should also compare the value of coercive field to the kink point of ρ_{xx} (H_{c1} and H_{c2})*

Reply: The reviewer suggests us to provide the M - H curves and comparing the coercive field of the kink features observed in magnetoresistance. Exactly following the suggestion, we provide the M - H curves of the samples with different x values in Fig. R2 below. Similar to the magnetoresistance, two kinds of kink features can also

be clearly observed in the M - H curves (marked by the arrows and triangles in Fig. R2(a)). The values of the coercive field of each kink point (H_{c1} and H_{c2}) and the x dependence evolution are consistent with the ones in ρ_{xx} data, as seen in Fig. R2(b). In the revised manuscript, we separate the thin film device transport data from the original Fig. 3 and provide M - H curves and the extracted H_{c1} , H_{c2} in Fig. 3b and 3d in the revised manuscript, respectively.

Fig. R2. (a) Magnetization versus B field of samples with different x values. (b) Comparison of the corresponding critical field for the kinks (H_{c1} , H_{c2}) extracted from the magnetoresistance and the M - H curves.

Comments: 4. *What's the sample thickness used for the transport and magnetic measurements? As $MnBi_2Te_4$ is an antiferromagnetic TI composed of ferromagnetic SLs with neighboring SLs coupled antiferromagnetically, there should be oscillation in magnetic properties as the thickness changes between even and odd SLs). Except for the 9 SL and 12 SL, the authors should show more data for the other thickness.*

Reply: The reviewer asked the thickness of the sample for transport and magnetic measurements. We are sorry that we didn't make a clear explanation in the original manuscript. The transport data displayed in Figs. 2d-2f and 3b in the original manuscript were taken from the plate-like bulk crystals with the thickness of 50 ~ 200 μm , which are cleaved by a knife and the size of these plates are 0.5 to 2 mm. In

samples with this kind of thickness, only bulk crystal properties can be detected. In the magnetic measurement, we use a vibrating sample magnetometer (VSM) to measure the magnetization of the samples. In order to obtain better signals, the samples for VSM measurement are much bigger and thicker, with about 0.5 ~ 1 mm in thickness and 2 ~ 3 mm in side length. In the revised manuscript, we make a clear explanation of the size of the bulk samples used in transport measurement in the Methods session. A detailed explanation of the magnetic measurement and the thin device transport measurement, including the equipment we used, the typical size of the samples and the fabrication process are also amended into the Methods session.

Fig. R3. (a) Hall resistance measured at 2 K in the $x = 0.1$ thin devices with different thickness. (b) The thickness dependent coercive fields extracted from panel a.

The reviewer also suggests us to provide more data for the samples with other thickness. We agree with the reviewer that this point is important and thank for the kind advice. Following the advice, we further fabricate thin films devices with various thicknesses from 5 to 9 SLs. The transport data of Hall resistance at 2K are provided

in Fig. R3(a) above. One can see that the devices with odd SLs (5, 7, 9 SLs) show obvious AHE with coercive field about 0.5 to 1 T. Though the hysteresis loops can also be observed in samples with even SLs (6, 8 SLs), the coercive fields of these samples are much smaller than the ones with odd SLs and the remnant magnetic signals are believed to result from the canting or disorder of the AFM configurations.³ Fig. R3(b) conclude the coercive field for devices with different thickness and the expected oscillation between even and odd SLs is clearly demonstrated. Since plenty of new transport data on thin devices are amended, also for avoiding confusion with the measurement on bulk samples, we separate the thin film device transport data from the original Fig. 3 and reorganize these data into Fig. 4 of the revised manuscript. Fig. R3 and the related discussion above are also provided in the manuscript.

As shown above, we have clarified all raised questions. We thank the reviewer for the detailed instructions, which help us to improve our manuscript greatly. We believe the manuscript has been improved and cordially wish that our revised manuscript is acceptable to *Nature Communications*.

Reference:

1. Gu, G. et al. Asperomagnetic order in diluted magnetic semiconductor (Ba,Na)(Zn,Mn)₂As₂. *Appl. Phys. Lett.* **112**, 032402 (2018).
2. Huat Lee, S. et al. Spin scattering and noncollinear spin structure-induced intrinsic anomalous Hall effect in antiferromagnetic topological insulator MnBi₂Te₄. Preprint at <https://arxiv.org/abs/1812.00339> (2018).
3. Gong, Y. et al. Experimental realization of an intrinsic magnetic topological insulator. *Chinese Physics Letters*. **36**, 076801 (2019).

Reply to Reviewer #2

The authors report on transport properties of magnetic topological insulator (MTI) $Mn(Sb,Bi)_2Te_4$ exfoliated single crystal and discuss a topological transition to non-TI by further Sb-doping. The Sb-doping to tune the Fermi level in the bulk band gap is firstly done in this compound. I found the work is technically sound and is an appropriate demonstration for the exploration of an intrinsic MTI phase for $MnBi_2Te_4$. The authors may need to address the following concerns before the manuscript could be considered for publication.

Reply: First of all, we are grateful to the reviewer for his (her) positive comments on our manuscript. We also thank the reviewer for the detailed instructions and the constructive suggestions, which help us to improve our manuscript greatly. According to the advices, the relevant part in the manuscript is amended and the manuscript has been improved. The questions raised by the reviewer are answered point by point as follows.

Comments: 1) *The authors achieved a p-n transition by Sb-doping to $MnBi_2Te_4$. According to the theory, the EF-tuned samples with x between 0.25 and 0.35 should show the Axion insulator state or the QAH state depending on the layer number of the sample. Namely, both σ_{xx} and σ_{xy} should be zero in the former case and $\sigma_{xx} = 0$ and σ_{xy} should be quantized to h/e^2 in the latter case. However, the transport data in Figs. 2d and 3b look far from those quantized states. The authors should mention the reason.*

Reply: The reviewer asked the reason why the transport data in Figs. 2d and 3b look far from quantization. Here, we answer this question on two aspects.

On one hand, we fully agree with the reviewer that the quantized transport is important and it is the ultimate goal of the researches on these MTI materials. According to the earlier research experience on topological materials, it is a long and hard course from materials to devices. For example, from the discovery of TIs in

Bi₂Se₃ family in 2009 to realizing the quantum Hall effect in surface transport dominated device in 2014,¹⁻² and from the theoretical prediction of QAHE in magnetic doped TIs in 2010 to the first experimental realization in 2013.³⁻⁴ In general, years of studies with multiple developmental stages are needed to achieve the topological quantum effects in devices, including theoretical predictions, material growth and initial experimental measurements, material optimizations, device fabrications and device optimizations, etc., each of which is a milestone benefiting the whole process from materials to devices. The concept of intrinsic MTI has been put forward just for several months. Though significant experimental progress has been achieved recently and some initial quantized transport phenomena have been realized, there is still a long way for the final realization of QAHE under zero B field and high temperature. Our results demonstrated in this manuscript are focused on the first step of material optimizations. We believe our results are helpful for enlightening further investigations and finally achieving long-expected quantized transport in these materials in the future.

On another hand, the transport data in Fig. 2d and 3b was taken from plate-like bulk crystals with the thickness of 50 ~ 200 μm . According to the previous reports, it is so far unable to completely suppress the bulk transport contribution in large size of bulk TI crystals, even in those greatly optimized TIs such as BiSbTeSe₂ and Sn-Bi_{1.1}Sb_{0.9}Te₂S.^{2,5-7} Though surface transport is dominated in thin film devices^{2,5}, the bulk contribution in these materials with hundreds of micrometers in thickness is still more than 80%.⁷ Similarly, in our work, although Sb substitution successfully suppresses the bulk carrier density comparing with MnBi₂Te₄, there are still considerable bulk contributions in the bulk crystals. We believe this is another reason why quantized transport is unable to be observed in these crystals. We are sorry that we didn't make a clear explanation of the size of the sample in the original manuscript. Thanks for the reviewer's comments, we provide more detailed information about the sample size and thickness in the main text and the Method session of the revised manuscript.

Comments: 2) Figure 3d and 3e: The authors show the presence and absence of hysteresis in R_{yx} depending on the layer numbers of SL. I have questions on the experiment. How homogeneous the layer numbers in the samples? I do not think the sample is completely flat. Then, how the layer numbers are determined? In addition, the authors should provide more detail information about the experiment such as a typical size, a photograph and the fabrication process of the sample.

Reply: The reviewer wonders about the flatness of our devices. We agree with the reviewer that it is important to the device transport measurement. In fact, when using a scotch-tape method to mechanically exfoliate the $\text{Mn}(\text{Sb,Bi})_2\text{Te}_4$ crystals, thin films with various thicknesses and surface morphologies can be achieved. Confirming by an AFM, we find that the thickness of some films is uniform, while plenty of steps can be observed in some others films. Fig. R1 shows the AFM data of several typical thin films we obtained. One can see the area of uniform thickness is large enough for the next step of device fabrication, and the layer number of each film is simply determined by $H_{\text{film}}/H_{\text{SL}}$ where H_{film} is the film thickness and H_{SL} is the height of one septuple layer of MnBi_2Te_4 (1.36 nm).

Fig. R1. AFM mapping data (a-c) and the corresponding linear cut profiles (d-f) of typical thin films of mechanically exfoliated $\text{Mn}(\text{Sb,Bi})_2\text{Te}_4$.

The reviewer also suggests us to provide more detail information about the thin

film devices transport measurement. We thank the reviewer for the kind suggestions. Following the reviewer's advice, we provide an AFM photograph of a typical 8 SL sample (Fig. R1c and 1f above) and an optical photograph of the corresponding fabricated thin device in Fig. 4a-c of the revised manuscript. The detailed fabrication process of the thin devices is also provided in the Methods session of the revised manuscript.

Comments: 3) *Related question: what is the layer number of the sample in Figs. 2d-2f? In particular, a ferromagnetic hysteresis is observed?*

Reply: As discussed above in the reply of comment 1, the transport data displayed in Figs. 2d-2f were taken from the plate-like bulk crystals with the thickness of 50 ~ 200 μm , which are cleaved by a knife and the size of these plates are 0.5 to 2 mm. The bulk MnBi_2Te_4 material is a compensated interlayer antiferromagnet since the magnetization direction of the neighboring septuple layers is opposite. Therefore, ferromagnetic hysteresis cannot be observed in the bulk MnBi_2Te_4 crystals with the thickness in micrometer scale.

We are sorry that we didn't provide detailed information about the transport measurement and confused the reviewer. We appreciate the reviewer for the reminding. In order to avoid confusion between the measurement on bulk samples and thin film devices, we separate the thin film device transport data from the original Fig. 3 and reorganize them into Fig. 4 of the revised manuscript. The detailed information about the measurement such as sample size and thickness are also provided in the Methods session of the revised manuscript.

Comments: 4) *Figure 4e: The size of the band gap changes by increasing Sb due to the reduction of spin-orbit interaction. What is the origin of the peak at $x = 0.9$? Why it shows a non-monotonic behavior?*

Reply: The reviewer asks why there is a peak for the calculated bandgap at $x = 0.9$. Thanks for the reviewer's comment. Here we provide more detailed explanations. We

think that the origin of this peak is the competition of two effects. The first is the spin orbit coupling (SOC) effect. We know that the SOC reduces with increasing Sb. The energy gap expects to enlarge by reducing the SOC. In Fig. R2(d), we calculate the energy gap with different SOC for MnBi_2Te_4 , which really presents that the energy gap becomes larger with reducing the SOC. The second is the chemical bonding effect. To see this effect, we calculate the band structures of MnSb_2Te_4 and MnBi_2Te_4 by setting the SOC to be zero, (or saying without the SOC), respectively, shown in Fig. R2(a)-(b). We can see that the energy gap of MnSb_2Te_4 is around 0.3 eV which is much smaller than that of MnBi_2Te_4 (around 0.6 eV). Therefore, the energy gap reduces when increasing Sb, schematically seen in Fig. R2(c). So, the energy gap depends on these two effects when increasing Sb. The reducing SOC enlarges the energy gap, but the chemical effect reduce the energy gap, when increasing the Sb in $\text{Mn}(\text{Sb},\text{Bi})_2\text{Te}_4$, schematically seen in Fig. R2(e). Finally, the peak of the energy gap appears at $x = 0.9$, seen in Fig. R2(f). Thanks for the reminding of the reviewer, we add this explanation and Figure R2 into the supplementary information.

Fig. R2. (a-b) The band structure of MnSb_2Te_4 without SOC. (b) The band structure of MnBi_2Te_4 without SOC. (c) The schematic of the chemical effect on

the energy gap from MnBi_2Te_4 to MnSb_2Te_4 . (d) The energy gap on the SOC effect for MnBi_2Te_4 . (e) Schematically combine the SOC effect and the chemical effect. The green line is the schematic energy gap on both SOC and chemical effects. A peak of the energy gap appears. (f) The calculated energy gap for $\text{Mn}(\text{Sb}_x\text{Bi}_{1-x})_2\text{Te}_4$.

Comments: *5) Although the authors claim that a topological phase transition lies at $x = 0.55$ from the band calculation, the samples shown in the manuscript are only $x < 0.5$. They should discuss the phase transition by providing ARPES or transport data for heavily Sb-doped samples.*

Reply: The reviewer suggests us to provide ARPES or transport data for heavily Sb-doped samples to discuss about the topological phase transition. We thank for the reviewer's kind suggestion. Unfortunately, from our ARPES and transport result in samples $x < 0.5$, we speculate that for the heavily Sb-doping MnBi_2Te_4 samples, the Fermi level would be located in the bulk valence band and far away from the Dirac point and the sample would be heavily p-doped. We also find a recent experimental report (*arXiv: 1905.00400*) in which the heavily Sb-doped samples have been investigated, confirming the heavily p-doped transport property. According to this arXiv report, there are no features in transport of heavily Sb-doped samples which reveals the topological phase transition. It is known that for transport measurement, only the carriers near the Fermi level contribute to the transport. In heavily p-doped samples, the transport is dominated by the bulk hole carriers and is hard to identify whether these samples are topological trivial or nontrivial. Similarly, for ARPES measurement, only the band structure below the Fermi level is detectable, thus for the heavily p-doped samples in which the Fermi level is in the bulk valence band, the features related to the band topology such as whether the surface states existing, or whether the bulk bandgap closing and reopening, are all above the Fermi level and unable to be detected. Therefore, according to the recent report, we believe it is unable to experimentally identify the topological nature in heavily Sb-doped samples by either ARPES or transport. Thanks for the reminding of the reviewer, we also cite this

recent arXiv report in our revised manuscript as a reference.

Comments: 6) *Related to the above, the authors mention in Fig. 3b that “We therefore believe that the magnetic transition point from AFM to paramagnetism in Mn(Sb,Bi)₂Te₄ lies outside the range studied here”. What is the evidence of the magnetic transition to paramagnetism by Sb-doping? The authors should provide a reference or an experimental data.*

Reply: Thanks for the reviewer’s reminding. We believe the reviewer misunderstands our meaning here. We are sorry for the inappropriate expression of this sentence which is easy to be misunderstood. In fact, we would like to express that although the AFM exchange coupling is reduced by Sb substituting, there is still no magnetic transition from AFM to paramagnetism within the range of $x < 0.5$ in Mn(Sb_xBi_{1-x})₂Te₄. In fact, in the recent arXiv report mentioned above (*arXiv:1905.00400*), the magnetism of MnSb₂Te₄ has been investigated. Though the Néel temperature decreases to 19 K, the spontaneous AFM order in MnSb₂Te₄ still maintains. That is to say, in the whole range of $x = 0$ to 1, there is no magnetic transition from AFM to paramagnetism in Mn(Sb_xBi_{1-x})₂Te₄. Thanks for the mention from the reviewer. This sentence has been amended in the revised manuscript as: “We therefore believe that there is no magnetic transition from AFM to paramagnetism and the spontaneous magnetization is maintained in Mn(Sb_xBi_{1-x})₂Te₄ within the range of $x < 0.5$. In fact, spontaneous AFM order still maintains when $x = 1$, i. e. in pure MnSb₂Te₄.⁸”

Comments: 7) *Figure 2b: Because all the EDX data are overlapped and are shown in similar colors, it is quite hard to see the composition dependence. The authors should separate the data into subpanels.*

Reply: We apologize for the low-contrast figure in the original manuscript. Thanks for the reviewer’s advice. We adjust the colors of Figure 2b in the main text, and each EDS curve is shifted for clearness, the modified figure is shown in Fig. R3 below. We thank the reviewer for this useful suggestion to improve our manuscript.

Fig. R3. EDS data after modification.

Comments: 8) Similarly, colors of traces in Figs. 2d and 2f should be revised. In addition, a new figure should be provided that shows a composition x dependence of carrier density.

Reply: We thank the reviewer for the kind suggestion. Following the reviewer's advice, the colors of the curves in Fig. 2(d)-(g) and Fig. 3(a)-(c) have also been amended in the revised manuscript, using the same color scale of Fig. R2 shown above. The reviewer also suggests us to provide a figure showing a composition x dependence of carrier density. Following this nice suggestion, we calculate the carrier density of each sample by the Hall coefficient extracted from Fig. 2e in the revised manuscript and provide this figure in Fig. 2f of the main text (Fig. R4 below).

Fig. R4. Carrier density in samples with different x values.

To summarize, we sincerely thank the reviewer for the detailed instructions. Following his/her suggestions and comments, we provide more detailed information about our experimental measurements, which helps the readers to better understand our work. Some inappropriate expressions and low-contrast figures are also amended. We think that all the suggestions and comments have been properly dealt in this revision, and the manuscript has been improved. We cordially wish the present version of the manuscript is satisfying for the acceptance.

Reference:

1. Zhang, H. et al. Topological insulators in Bi_2Se_3 , Bi_2Te_3 and Sb_2Te_3 with a single Dirac cone on the surface. *Nature Physics*. **5**, 438 (2009).
2. Xu, Y. et al. Observation of topological surface state quantum Hall effect in an intrinsic three-dimensional topological insulator. *Nature Physics*. **10**, 956 (2014).
3. Yu, R. et al. Quantized Anomalous Hall Effect in Magnetic Topological Insulators. *Science*. **329**, 61-64 (2010).
4. Chang, C.-Z. et al. Experimental Observation of the Quantum Anomalous Hall Effect in a Magnetic Topological Insulator. *Science*. **340**, 167-170 (2013).
5. Zhang, S. et al. Anomalous quantization trajectory and parity anomaly in Co cluster decorated BiSbTeSe_2 nanodevices. *Nature Communications*. **8**, 977 (2017).
6. Kushwaha, S. et al. Sn-doped $\text{Bi}_{1.1}\text{Sb}_{0.9}\text{Te}_2\text{S}$ bulk crystal topological insulator with excellent properties. *Nature Communications*. **7**, 11456 (2016).
7. Wu, B. et al. Oscillating planar Hall response in bulk crystal of topological insulator Sn doped $\text{Bi}_{1.1}\text{Sb}_{0.9}\text{Te}_2\text{S}$. *Appl. Phys. Lett.* **113**, 011902 (2018).
8. Yan, J.-Q. et al. Evolution of structural, magnetic and transport properties in $\text{MnBi}_{2-x}\text{Sb}_x\text{Te}_4$. Preprint at <https://arxiv.org/abs/1905.00400> (2019).

Reply to Reviewer #3

The authors studied a series of $Mn(SbxBi_{1-x})_2Te_4$ samples, by angle-resolved photoemission spectroscopy, transport measurements and DFT calculations. ARPES and transport data show a quite convincing n-p transition, and transport data show AFM magnetic transition in all studied samples ($x \leq 0.5$). Together with DFT calculations, the authors concluded that an ideal MTI zone can be reached by tuning the Sb/Bi ration, where topological nature, bulk carrier suppression, and spontaneous magnetization can be satisfied at the same time, and $x \sim 0.3$ is likely the best composition. These evidences are comprehensive.

Reply: First of all, we thank the reviewer for reading our manuscript and believing that most of the evidences in our work are comprehensive. The reviewer gives us several detailed instructions of comments and suggestions, which help us to improve our manuscript greatly. Following his/her criticisms and recommendations, we revise our manuscript and provide point-by-point answers to the reviewer's comments as follows.

Comments: *However, the main purpose of this study is to find the ideal MTI, and the authors concluded it is $x \sim 0.3$. But they showed almost no data for $x \sim 0.3$. Some of the key evidences (topological surface states and QAH) are carried out only on $x = 0$ or $x = 0.1$ samples, of which similar results were already posted on arXiv a few months ago (The gapped topological surface states for $x = 0$ are also observed in arXiv 1809.07389(Ref.33), 1809.07926(Ref.34), 1812.00339(Ref.35), 1903.11826. The QAH for $x = 0.1$ thin-layer sample is similar to the one reported in arXiv 1809.07926(Ref.34) for $x = 0$). In my opinion, to reduce the overlap with the previous papers and to further support the conclusion, some data on $x \sim 0.3$ (pure topological states, or better QAH, etc) is necessary.*

Reply: The reviewer comments that more data on $x \sim 0.3$ thin film samples should be provided. We fully agree with the reviewer that the transport data on thin film samples

with optimized Fermi level is important. Following the reviewer's suggestion, we carried out further investigation in these materials in recent months. During the continuous experiments, we are surprised to recognize an important factor that has not been noticed before, it is that the thin film device fabrication process would cause the Fermi level shifting downwards. Thus, the optimized x value for thin devices is smaller than ~ 0.3 . The mechanism of this kind of unexpected Fermi level shifting is still unclear and it seems a general phenomenon according to several experimental reports on MnBi_2Te_4 devices recently.¹⁻³ P-type contaminant introduced during device fabrication is a possible reason, but this phenomenon was rarely reported in the previous studies of TI thin devices using the similar fabricating technic. Thus, maybe some intrinsic properties of MnBi_2Te_4 crystals are related, which are worth for further studies. We thank the reviewer for the insightful suggestions, which help us to realize the unexpected Fermi level shifting in thin devices and to improve our manuscript greatly. The detailed discussions on the Fermi level shifting in thin devices are as follows.

Fig. R1. Resistance versus gate voltage in $\text{Mn}(\text{Sb}_x\text{Bi}_{1-x})_2\text{Te}_4$ thin film devices with different x values. (a)-(c) $x = 0$. (d)-(g) $x = 0.1$. (h) $x = 0.25$. (i) $x = 0.3$.

According to the plenty of $\text{Mn}(\text{Sb},\text{Bi})_2\text{Te}_4$ devices we fabricated and measured in the last several months, we find that though the specific shifted distance of Fermi level in each device is strongly sample-dependent, most of the pure MnBi_2Te_4 thin films are still heavily n-doped and hard to tune the samples to the electrical neutral point by applying gate voltage, as shown in Fig. R1(a)-(c). Thus, we believe that the carrier moderation by antimony substituting is still essential in device transport studies of $\text{Mn}(\text{Sb},\text{Bi})_2\text{Te}_4$. For thin films with $x = 0.1$. The situation changes enormously as shown in the resistance versus gate voltage curves displayed in Fig. R1(d)-(g). One can see that the electrical neutral point can be achieved under a moderate gate voltage in most of the $x = 0.1$ thin film samples (marked by the pink shaded areas in Fig. R1(d)-(g)). Though some of the $x = 0.1$ devices are slightly n-doped while some others are p-doped because of the sample dependence of Fermi level shifting, we consider that this rate of Sb substituting ($x \sim 0.1$) is appropriate for thin devices. Further measurement in thin film devices with $x = 0.25$ and $x = 0.3$ also confirm the Fermi level shifting in $\text{Mn}(\text{Sb},\text{Bi})_2\text{Te}_4$ that they are heavily p-doped and hard to achieve the electrical neutral point by gate tuning (Fig. R1(h)-(i)).

As discussed above, for thin film devices, $x \sim 0.1$ is a more appropriate substituting ratio than $x \sim 0.3$. We thank the reviewer for the detailed instructions, which guide us to further investigate the thin film devices and find more new results. We discuss the thin film devices in more detail in the revised manuscript. We provide more detailed devices transport data and re-organize them into Fig. 4 of the revised manuscript. The phase diagram in Fig. 5e (4e in the original manuscript) is also amended to add the information of the optimized range of x for thin film devices. The R - V_g curves for different $\text{Mn}(\text{Sb}_x\text{Bi}_{1-x})_2\text{Te}_4$ thin film devices displayed in Fig. R1 above are also provided in the Supplementary Information.

Comments: *Similar ARPES spectra were already measured by previous papers. Especially, Fig.1 is similar to Fig.3 in arXiv 1809.07389(Ref.33), though experimental quality in Ref.33 seems to be better. The x dependence of ARPES in*

Fig.2 is new, but spectra are not clear. The ARPES data on $x \sim 0.3$ and its topological surface state is necessary.

Reply: Thanks for the reminding of the reviewer, following the guidance of the reviewer, we carried out more systematic ARPES measurement on the MnBi_2Te_4 system. After collecting the ARPES spectra more carefully and precisely, we discover new results on the surface states dispersions that there is actually no observable gap opening at the Dirac point detected by ARPES even under the Néel temperature of AFM transition (8 K). This result is surprised and distinct with what we thought previously. We make intense modifications on the related part in the revised manuscript and we sincerely thank the guidance from the reviewer. Below, we will discuss on these new results in detail.

In the new round of ARPES measurement, the condition of the equipment is better and allows us to use a lower temperature of 8 K to collect the spectra, and the time spending for signal integration is longer to collect the spectra with better quality. Especially, smaller angle step is used when looking for the position of Γ point. Since the Fermi velocity of the surface states in MnBi_2Te_4 is relatively large. A slightly deviation from the Γ point would cause considerable misreading of the gap size. The newly-collected ARPES data is displayed in Fig. R2, one can see that from the zoom-in view (Fig. R2(b)), the linear dispersion of both the upper and lower part of the surface Dirac cone seems well maintained. Combing with the EDC and MDC displayed in Fig. R2(c)-(d), there is still finite spectral weight near the Dirac point (bold line in Fig. R2(c)), and the intensity of EDC near the Dirac point changes almost linearly with no obvious ‘U’ shaped gap opening features. Considering the resolution limit of ARPES, we believe that the gap opening size in real synthesized MnBi_2Te_4 crystals is much smaller than what we expected previously (~ 50 meV) and cannot be identified by our ARPES.⁴ The gap opening feature in the previous measurement may be an artifact caused by the slight deviation from $k_x = k_y = 0$ because of the large Fermi velocity of the surface states.⁴ Another possible reason for observing the gap opening is that the intensity of the surface state is quite low and sensitive to the photon energies.⁴

Fig. R2. (a) ARPES spectrum of MnBi_2Te_4 collected at $h\nu = 7.25$ eV and $T = 8$ K. (b) A zoom-in view of ARPES spectrum near the Dirac point. (c) The MDCs extracted in the area marked by the dashed rectangular box in (b). (d) The EDC extracted at $k_{\parallel} = 0$ in (a).

In fact, we also find a recent experimental report discussing about thin film MnBi_2Te_4 devices transport. In fabricated MnBi_2Te_4 devices, the measured exchange gap at $B = 12$ T is only $\Delta = 21$ K detected by transport measurement which is much smaller than the one expected by calculation, and the exchange gap is even smaller under zero magnetic field than under $B = 12$ T.¹ The transport result in this report is contrast to the big gap opening with dozens of meV observed in our previous ARPES measurement. Thus, our new ARPES result with no observable bandgap at the Dirac point provides a new understanding on this material and reflects a more realistic picture of the surface states in real synthesized MnBi_2Te_4 crystals.

In addition, the reviewer also points out the poor quality of the ARPES data in samples with different x values and requests the ARPES data of the surface states on $x = 0.3$. We fully agree with the reviewer that the high-quality ARPES data is important. According to the request, we further precisely optimize the crystal growth parameters and achieve single crystals with $x = 0.2, 0.3$ and 0.4 with larger crystal sizes and flatter surfaces, which are better for ARPES measurement. Then, we use the ARPES equipment located at the Shanghai Synchrotron Radiation Facility to re-collect the

ARPES spectra of these crystals. The collection of the photoelectron is much efficient, and the intensity of the spectra we get is almost two orders of magnitude larger than the previous spectra displayed in Fig. 2c in the original manuscript. The new ARPES data are displayed in Fig. 2c of the revised manuscript (Fig. R3 below). The Fermi level tuning by x values can be clearly identified in these new data. For the sample of $x = 0.3$, due to the higher-quality data, the Fermi level of this sample can be observed much clearer. One can see that the Fermi level is just above the bulk valence band, and the surface state in the band gap is just unable to be detected by ARPES.

Fig. R3. ARPES spectra measured at $h\nu = 14$ eV for the samples with different antimony substituting from $x = 0$ to 0.4.

Comments: *There is no clear evidence of topological surface state in ARPES. Spin resolved ARPES is powerful in order to reveal the topological surface state. There have been already similar ARPES data in the previous papers. The authors should try spin resolved ARPES as a new experiment.*

Reply: Thanks for the reviewer's reminding. We strongly agree with the reviewer that the spin ARPES is a powerful technique to identify the surface state by analyzing the spin-momentum lock texture. However, the intensity of the surface states in MnBi_2Te_4 is sensitive to the photon energy and can be only clearly detected at $h\nu \sim 7$ eV. We feel very sorry that a spin ARPES equipment which can work under this range of photon

energy is beyond the experimental resource we can obtain. Thus, unfortunately, we are unable to satisfy the reviewer's request for spin ARPES measurement. Alternatively, following the reviewer's suggestion, we try to find other evidence of the topological surface states. As discussed above, we re-collect the ARPES data of MnBi_2Te_4 under 7.25 eV more precisely, and data with better quality is achieved. We find that there is no clear gap opening feature of the surface states and the cone shaped feature with linear band dispersions is observed. We believe this is also a strong evidence to prove the surface Dirac cone nature of the linear band observed by our ARPES measurement.

We thank the reviewer for the comments. These new ARPES data provide us distinct understandings on surface band dispersions comparing with what we previous thought in the original manuscript. Thanks for the reviewer's insightful advice to improve our revised manuscript.

Comments: (1). *For the ARPES kz data on $x = 0$ (Fig. 1g), is there any surface state in the second derivative plot? arXiv 1809.07389(Ref.33) observed both bulk bands and surface bands with $h\nu = 9$ eV in the second derivative. If not, what is the possible difference?*

Reply: We thank the reviewer for the comments. Following the reviewer's question, we further investigate the ARPES data under 9 eV. Fig R4(a) displays the raw data of the ARPES spectrum on $x = 0$ sample. When comparing with the data in the reference mentioned by the reviewer (*arXiv 1809.07389*, Figure 3c), one may notice that in both two figures from us and the reference, the signal intensity from the surface states is very weak while the intensity of the bulk bands is much stronger. The band dispersions achieved by two groups are qualitatively consistent with each other. The corresponding second derivative spectrum of our ARPES data is also plotted in Fig. R4(b), comparing with Figure 3d in the reference, the similar dispersions from both the upper part of the surface Dirac cone (yellow dashed line) and the bulk conduction band (red dashed line) can be identified. However, the quality of our data is not good enough to identify the lower part of the surface Dirac cone. The possible reason is that

the time for collecting this spectrum is too short to achieve enough intensity of signals from surface states, because the surface states intensity is weak at 9 eV and much longer time for signal integral is needed. In addition, from our ARPES measurement, we notice that the surface states intensity changes dramatically near the photon energy of ~ 9 eV. Thus, slight disagreement on the photon energy may also cause the relatively big difference on the intensity of the surface states. We believe these two points mentioned above are the possible reasons causing the difference between the two spectra from two research groups. Anyway, though the quality of the ARPES spectrum under 9 eV in the reference is better than ours, these two results are still qualitatively consistent with each other.

Fig. R4. (a) The ARPES spectrum of MnBi₂Te₄. (b) The corresponding second derivative spectrum.

Comments: (2). According to the calculation (Fig. 1f), the bulk bands show very small k_z dispersion (possibly due to the large lattice constant c). There should be no obvious k_z change for bulk bands in ARPES. In Fig. 1(g), why the data from $h\nu = 13.75$ eV is quite different from others?

Reply: The reviewer feels confused about the large k_z dispersion detected by ARPES and the unique spectra at $h\nu = 13.75$ eV comparing with other photon energies. We thank the reviewer for the comments. Following the comment, we carry out further first-principle calculations and provide more data on the ARPES measurement. Here,

we respond the reviewer from two aspects as follows.

Firstly, from our experimental results, we emphasize that k_z dispersion is exactly the reason why the spectra at $h\nu = 13.75$ eV looks different from the ones under other photon energies. For more detailed information of the ARPES measurement, please look at Fig. R5 displayed below. We provide more ARPES data under different photon energies from 11 eV to 17 eV. The EDC at $k_{\parallel} = 0$ (solid lines) and the corresponding peak positions of the conduction band minimum and the valance band maximum (dashed lines) are also displayed. One can clearly see the evolution of the conduction and valance band under different photon energies. When $h\nu = 13.75$ eV, the detected bulk bandgap is smaller than other photon energies in Fig. 1(g) (9, 11, 15, eV). This is the reason why the data at $h\nu = 13.75$ eV in Fig. 1g of the main text looks unique. From Fig. R5 we find that k_z dispersions of valance band is about 0.05 eV at $k_{\parallel} = 0$, which is much obvious than the relatively small k_z dispersion of the conduction band.

Fig. R5. Dispersions of MnBi_2Te_4 under different photon energies from 11 eV to 17 eV. The corresponding EDC at $k_{\parallel} = 0$ is plotted in each panel, and the position of the conduction band minimum and the valance band maximum are also marked by the dashed lines.

Secondly, from a theoretical point of view, after doing more first-principles calculations on this issue, we have to say that the prediction of the exact energy dispersion is really beyond the DFT calculations. At first, we calculate the band structures on different Coulomb U values with the PBE+ U functional for MnBi_2Te_4 , seen Fig. R6(a)-(c). The finite U values are used to open a full energy gap and make

the bands become flat along k_z direction (Γ -Z direction). We also see that the flat energy dispersion along k_z direction is not sensitive with the finite U values. Secondly, we carried out the same calculations with the LDA+U functional, seen in Fig. R6(d)-(f). We have the essentially same conclusion. For example, the finite U values open a full energy gap and make the bands be flat along k_z direction. But the energy dispersion along k_z direction seems not to be the same flat as that of PBE+U. Based on these calculations, we conclude that though the calculations are qualitatively the same for both PBE+U and LDA+U (e.g. the energy gap, the topological property), there are still some quantitative differences. The exact prediction of the energy dispersion is beyond the DFT functionals.

Fig. R6. The band structures of MnBi_2Te_4 . (a)-(c) The bands structures are calculated with the PBE+U functional. The U value is 0 eV in (a), 2 eV in (b) and 3 eV in (c). (d)-(f) The band structures are calculated with the LDA+U functional. The U value is 0 eV in (d), 2 eV in (e) and 3 eV in (f). We can see that the finite U values make a full energy gap. The band structures seem no essential difference from PBE+U and LDA+U functionals.

Comments: (3). In Fig.2(c) panel v ($x = 0.25$), if the red line is the surface band and the black line is the bulk band, the bulk band gap is ~ 250 meV. While for $x = 0$, it is less than 200 meV from Fig. 1(d). The bulk gap should reduce with Sb substitution. How to explain this contradiction?

Reply: We thank the reviewer for the comments. As discussed above, higher-quality ARPES spectra of the samples with $x = 0$ and 0.2 are obtained, as shown in Fig. R7. According to the EDC extracted at $k_{//} = 0$ from our new ARPES data, the bulk bandgap of pure MnBi_2Te_4 and $\text{Mn}(\text{Sb}_{0.2}\text{Bi}_{0.8})_2\text{Te}_4$ are similar (~ 160 meV) as shown in Fig. R7 below, there are no observable difference between these two samples considering the resolution limit of our ARPES measurement (~ 10 meV). In the previous ARPES data with lower quality, a cone shaped band marked by the red dashed line in Fig. 2c(v) is considered to be the surface state, however, according to our new ARPES results. This band in Fig. 2c(v) in the previous manuscript is more likely to be the bulk valence band, thus the bulk bandgap measured in Fig. 2c(v) is also about ~ 160 meV, consistent with our new ARPES result. In general, it seems that the bulk bandgap does not change greatly in our ARPES measurement which is contrast to the theoretical calculation. One of the possible reasons is that the corresponding k_z value of the samples with $x = 0$ and $x = 0.2$ is different under the same photon energy and the k_z dispersions may affect the measurement of the gap value. The slight difference of the chemical potential in different area of the samples caused by disorders would also influence the estimating of the band gap. Another factor that the exact prediction of the energy dispersion is beyond the DFT functionals discussed above in the reply of the previous comment may also a possible reason. Anyway, thank for the reviewer's reminding, the exact reason why the detected gap value by ARPES is not consistent the theoretical calculation is still unclear and worthy for further investigations.

Fig. R7. Newly collected ARPES spectra for the samples with $x = 0$ and 0.2 at $h\nu = 14$ eV, $T = 11$ K.

Comments: (4). In Fig.3(d-e), why do the authors use R_{xy} for (d) and ΔR_{xy} for (e)? What is the definition of ΔR_{xy} ?

Reply: Thanks for the reminding of the reviewer. We are sorry that we didn't make a clear definition of ΔR_{xy} in the original manuscript. In Fig. 3d, R_{xy} represents the original Hall resistance of the device with thickness of 12 SL. While in Fig. 3e, because AHE signal can be observed in the device with 9 SL, we subtract the linear background contributed by the original Hall resistance from the raw data, to clearly display the anomalous Hall signal with a hysteresis loop. This is the reason we used a 'Δ' mark for the y axis of Fig. 3e in the original manuscript. In the revised manuscript, new transport data on thin devices has been provided and reorganized in Fig. 4 of the main text. The AHE loops of these new devices are clearer and we directly display the raw R_{xy} data in Fig. 4e, thus the unit of the y axis of this figure in the revised manuscript is R_{xy} without 'Δ' mark.

Comments: In summary, there is no strong support for the topological nature and the related QAH for the ideal MTI ($x \sim 0.3$). The data on the chemical potential shift and AF magnetization with Sb substitution are solid. But these solid data are not be enough for Nature Communications.

Reply: We thank the reviewer for reading our manuscript and giving us detailed

instructions, which help us to greatly improve our manuscript. Following the reviewer's insightful comments and suggestions, we further investigate the $\text{Mn}(\text{Sb,Bi})_2\text{Te}_4$ family in detail and systematically amended the experimental results on ARPES and transport measurement. More new results and new understanding are provided. We believe this manuscript has been greatly improved.

As a summary, our manuscript is the first study to investigate the possibility of Fermi level adjusting and carrier modulation in intrinsic MTIs. By various research techniques including ARPES, electrical transport, magnetic measurement and first-principle calculations, systematical investigations on the position of Fermi level, carrier type and density, magnetization and topological property of $\text{Mn}(\text{Sb,Bi})_2\text{Te}_4$ family have been carried on. Recently, the researches on intrinsic MTIs including MnBi_2Te_4 family are springing up rapidly and some exciting transport results have been reported, but there is still a long way for the final realization of high temperature QAHE which can be observed without the assistance of external magnetic field. We believe our results presented in this manuscript are helpful for enlightening further investigations and getting more fruitful results of intrinsic MTIs in the future. We thank the reviewer for the detailed instructions. We believe this version of manuscript has been improved following the the reviewer's advices and comments. We cordially wish the reviewer satisfies with our detailed replies above, and agrees the acceptance of the revised manuscript in *Nature Communications*.

References:

1. Deng, Y. et al. Magnetic-field-induced quantized anomalous Hall effect in intrinsic magnetic topological insulator MnBi_2Te_4 . Preprint at <https://arxiv.org/abs/1904.11468> (2019).
2. Liu, C. et al. Quantum phase transition from axion insulator to Chern insulator in MnBi_2Te_4 . Preprint at <https://arxiv.org/abs/1905.00715> (2019).
3. Cui, J. et al. Transport properties of thin flakes of the antiferromagnetic topological insulator MnBi_2Te_4 . *Phys. Rev. B.* **99**, 155125 (2019).
4. Discussed with Prof. Ke He from Tsinghua University.

REVIEWERS' COMMENTS:

Reviewer #1 (Remarks to the Author):

I think the authors have satisfactorily addressed my concerns. But I have new questions on the AHE data in Fig. 4(e). Why the AHE curves of even layers show such a strange shape? Why the AHE of 8 SL changes sign from negative to positive? The authors should give an explanation for this before publication.

Reviewer #2 (Remarks to the Author):

Because the questions/comments we asked are well addressed, we recommend the publication with no further comments for the authors.

Reviewer #3 (Remarks to the Author):

I am satisfied with this revision.

Reviewer #4 (Remarks to the Author):

I carefully read the revised manuscript by B. Chen, et al and the comments of the Reviewer 3. I personally think the authors have addressed most of the questions raised by the reviewer.

The biggest concern of the reviewer 3 is that the topological properties of the materials (especially the $x=0.3$ compound) are not sufficiently characterized. And the reviewer suggests to carry out spin-resolved ARPES or show transport results with quantized Hall resistance.

I partially agree with the reviewer. Fig. 2c clearly demonstrates the chemical potential shift with various x . However, it would be great, if the author can present better ARPES data with clear surface bands in the samples with $x = 0.2$ or 0.25 . These data will be strong evidence for the success of preserving the topological properties of the samples during the control of doping level. To obtain better signal of the surface bands, it might be helpful to select the photon energy at 7 eV rather than 14 eV.

On the other hand, I think that the manuscript has shown a good approach to get intrinsic magnetic topological insulator. The authors already demonstrate an efficient way to fine tune the chemical potential and magnetism of the 124 compound. I understand that it will be a long way to realize QAH in transport measurements. At the current stage, the results are enlightening and important.

One suggestion: The absence of gap opening at the Dirac point might also originate from the change of magnetic moments on the surface of Mn-124. The author may want to add this possibility to the main text.

Reviewer #1 (Remarks to the Author):

I think the authors have satisfactorily addressed my concerns. But I have new questions on the AHE data in Fig. 4(e). Why the AHE curves of even layers show such a strange shape? Why the AHE of 8 SL changes sign from negative to positive? The authors should give an explanation for this before publication.

Reply to Reviewer #1

First of all, we sincerely thank the reviewer for his/her careful reading on our manuscript, and we appreciate the reviewer's positive comments on our manuscript and replies! The reviewer pointed out new questions on our new anomalous Hall effect (AHE) data in Fig. 4(e).

For the strange shape of the AHE curves of even layers, we believe there are two possible origins. Firstly, we need to mention that for the data in Fig. 4(e), we do not substrate the background signals from the normal Hall contributions which means the total Hall resistance $R_{xy}(\text{total}) = R_{xy}(\text{normal}) + R_{xy}(\text{AHE})$ is displayed. In each device, a back gate voltage is applied to reach the maximum longitudinal resistance. Thus the $R_{xy}(\text{normal})$ may show a strongly nonlinear behavior due to the two components of carrier contributions, and there will be a strange shape of background in the measured $R_{xy}(\text{total})$. Secondly, the remnant AHE signals in samples with even layers are believed to result from the canting or disorder of the AFM configurations. The specific distributions of the magnetic moment in these samples are still unclear, possibly sample dependence due to the randomness of the disorder which is another explanation for the strange shape of the AHE curves in even layers.

For the AHE in samples with 8 SL, we actually cannot fully get the reviewer's meaning on the changed sign from negative to positive. Does the reviewer mean that

the slope of the R_{xy} background is negative in most samples but positive in 8 SL? If so, it can be understood by the reason discussed above that the slope is influenced by the normal Hall contributions and the slope could be positive or negative (or nonlinear) when the Fermi level is close to the charge neutral point. The reviewer may also mean that the direction of the hysteresis loop near zero field in 8 SL is opposite. Actually, according to a recent report (arXiv: 1905.04839), the reversal of AHE signal is an exotic phenomenon which has also been observed in MnBi_2Te_4 when tuning the Fermi level close to the bandgap. It is believed to come from the competition between the intrinsic Berry curvature and Dirac-gap enhanced extrinsic skew scattering in this material. Since the Fermi level is also close to the bandgap in our samples displayed in Fig. 4e, the reversal of AHE signals in our samples may come from the same reason.

Following the reviewer's advice, a new statement "*Since the Fermi level is close to the charge neutral point, the reversal of AHE signals in Fig. 4e may come from the competition between the intrinsic Berry curvature and Dirac-gap enhanced extrinsic skew scattering in this material.*" has been added into the manuscript.

To summarize, we thank the reviewer for the detailed instructions. We sincerely hope that the reviewer is satisfied with our replies. Thanks to his/her nice suggestions. We cordially wish the present version of the manuscript is satisfying for the acceptance.

Reviewer #2 (Remarks to the Author):

Because the questions/comments we asked are well addressed, we recommend the publication with no further comments for the authors.

Reply to Reviewer #2

We sincerely thank the reviewer for reading our manuscript and providing the detailed instructions to help us to improve our manuscript. We also appreciate the reviewer's recommendation for publication!

Reviewer #3 (Remarks to the Author):

I am satisfied with this revision.

Reply to Reviewer #3

We sincerely thank the reviewer for reading our manuscript and providing the detailed instructions to help us to improve our manuscript. We are pleased to see the reviewer is satisfied with our revisions. We also appreciate the reviewer's recommendation for publication!

Reviewer #4 (Remarks to the Author):

I carefully read the revised manuscript by B. Chen, et al and the comments of the Reviewer 3. I personally think the authors have addressed most of the questions raised by the reviewer.

The biggest concern of the reviewer 3 is that the topological properties of the materials (especially the $x=0.3$ compound) are not sufficiently characterized. And the reviewer suggests to carry out spin-resolved ARPES or show transport results with quantized Hall resistance.

I partially agree with the reviewer. Fig. 2c clear demonstrates the chemical potential shift with various x . However, it would be great, if the author can present better ARPES data with clear surface bands in the samples with $x = 0.2$ or 0.25 . These data will be strong evidence for the success of preserving the topological properties of the samples during the control of doping level. To obtain better signal of the surface bands, it might be helpful to select the photon energy at 7 eV rather than 14 eV.

On the other hand, I think that the manuscript has shown a good approach to get intrinsic magnetic topological insulator. The authors already demonstrate an efficient way to fine tune the chemical potential and magnetism of the 124 compound. I understand that it will be a long way to realize QAH in transport measurements. At the current stage, the results are enlightening and important.

One suggestion: The absence of gap opening at the Dirac point might also originate from the change of magnetic moments on the surface of Mn-124. The author may want to add this possibility to the main text.

Reply to Reviewer #4

I carefully read the revised manuscript by B. Chen, et al and the comments of the Reviewer 3. I personally think the authors have addressed most of the questions raised by the reviewer.

Reply: We thank the reviewer for reading our manuscript. We are grateful for the reviewer's positive comments on our works and replies!

Comments: *I partially agree with the reviewer. Fig. 2c clear demonstrates the chemical potential shift with various x . However, it would be great, if the author can present better ARPES data with clear surface bands in the samples with $x = 0.2$ or 0.25 . These data will be strong evidence for the success of preserving the topological properties of the samples during the control of doping level. To obtain better signal of the surface bands, it might be helpful to select the photon energy at 7 eV rather than 14 eV.*

Reply: The reviewer suggests us to provide some ARPES data of the doping ratio $x = 0.2$ or 0.25 around 7 eV to show the surface bands. Thanks for the suggestions and we fully agree with the reviewer's opinion. It is noteworthy that the intensity of the surface states in the collected spectra is sensitive to the incident photon energy. Only a small range of photon energy can be used to observe the surface dispersion in this family of material. For MnBi_2Te_4 , after plenty of trials by our group (and several other groups), the suitable range of photon energy is about $6\sim 7$ eV. But for the Sb doped samples, the suitable range seems to change. Fig. R1(a) displays the spectra of $x = 0.1$ sample collected under $h\nu = 7.25$ eV. Similar to the spectra of $x = 0$ sample in the main text, one can still distinguish the surface states and the bulk bands in this spectra, but the intensity of the surface band is much lower than the spectra of $x = 0$ sample and the linear dispersion of the two bands forming Dirac cone is not so clear as the $x = 0$ sample. When using 7.25 eV to collect the spectra of $x = 0.2$ sample, the surface

bands are difficult to observe under this photon energy and only bulk bands with a big bulk gap are detectable (Fig. R1(b)). Then we try to change the incident photon energy a little, using 8.4 eV (xenon lamp as the light source) to collect the spectra of $x = 0.2$ sample, as shown in Fig. R1(c). One can see that the spectrum changes greatly from 7.25 eV to 8.4 eV. This kind of big change unlikely comes from the bulk states since the k_z dispersion in this family of material is small, and the spectra in Fig. R1(c) is similar to the spectra of the sample with $x = 0.1$ and $x = 0$ collected at 7.25 eV (Fig. R1(a) below, and Fig. 1b in the main text). Thus, it is reasonable to identify it as a signature of the surface state in $x = 0.2$ samples, attributing to the increase of the intensity of the surface bands under $h\nu = 8.4$ eV.

Fig. R1. ARPES spectra measured at $h\nu = 7.25$ eV of (a) 10% and (b) 20% Sb doping. (c) ARPES image of $x = 0.2$ doping ratio under 8.4 eV photon energy.

We admit that the quality of the spectra in Fig. R1(c) is low and hard to distinguish the surface dispersions. We believe it is due to the relatively low luminous flux of the xenon lamp, and 8.4 eV may still not the optimal photon energy to observe the surface dispersion in $x = 0.2$. More detailed and systematical measurements based on the synchrotron radiation ARPES equipment is needed for the further investigation on the surface band dispersions in $x = 0.2$ samples. Unfortunately, due to the equipment maintenance recently and the limited machine-hour, it will take more than

three months for us to access the synchrotron radiation ARPES sources. However, we think the ARPES data of 8.4 eV is a convincing evidence to prove the existence of the surface states at $x = 0.2$ sample. We truly wish the existing data can meet the reviewer's requirements.

Comments: *On the other hand, I think that the manuscript has shown a good approach to get intrinsic magnetic topological insulator. The authors already demonstrate an efficient way to fine tune the chemical potential and magnetism of the 124 compound. I understand that it will be a long way to realize QAH in transport measurements. At the current stage, the results are enlightening and important.*

Reply: We appreciate the reviewer's positive comments on our manuscript!

Comments: *One suggestion: The absence of gap opening at the Dirac point might also originate from the change of magnetic moments on the surface of Mn-124. The author may want to add this possibility to the main text.*

Reply: We fully agree with the reviewer's suggestion for the gapless surface state. Thank for the nice suggestion. Following this advice, a new statement "*The contradiction between theoretical calculations and experimental measurements possibly comes from the complex magnetic moment distributions at the surface of the $MnBi_2Te_4$ crystals, where the moments may not be strictly arranged like the A-type AFM order in the bulk.*" has been added into the manuscript.

To summarize, we sincerely thank the reviewer for the detailed instructions. We think that all the suggestions and comments have been properly dealt in this revision. We cordially wish the present version of the manuscript is satisfying for the acceptance.